# Identifying metabolic reprogramming phenotypes with glycolysis-lipid metabolism discoordination and intercellular communication for lung adenocarcinoma metastasis

Xin Li[1,5], Lefan Tang[1,5], Jiaxing Deng[1], Xiuying Qi[2], Juxuan Zhang[1], Haitao Qi[1], Mengyue Li[1], Yixin Liu[3], Wenyuan Zhao[1], Yunyan Gu [1], Lishuang Qi [1✉] & Xia Li [1,4✉]

Tumor metastasis imposes metabolic requirements for escaping from primary tissues, producing vulnerability in treatment. This study aimed to explore the metabolic reprogramming relevant to lung adenocarcinoma (LUAD) metastasis and decode the underlying intercellular alterations. Using the gene expression profiles of 394 LUAD samples derived from The Cancer Genome Atlas (TCGA), we identified 11 metastasis-related metabolic genes involved in glycolysis and lipid metabolism, and defined three metabolic reprogramming phenotypes (MP-I, -II, and -III) using unsupervised clustering. MP-III with the highest glycolytic and lowest lipid metabolic levels exhibited the highest metastatic potency and poorest survival in TCGA and six independent cohorts totaling 1,235 samples. Genomic analyses showed that mutations in *TP53* and *KEAP1*, and deletions in *SETD2* and *PBRM1* might drive metabolic reprogramming in MP-III. Single-cell RNA-sequencing data from LUAD validated a metabolic evolutionary trajectory from normal to MP-II and MP-III, through MP-I. The further intercellular communications revealed that MP-III interacted uniquely with endothelial cells and fibroblasts in the ANGPTL pathway, and had stronger interactions with endothelial cells in the VEGF pathway. Herein, glycolysis-lipid dysregulation patterns suggested metabolic reprogramming phenotypes relevant to metastasis. Further insights into the oncogenic drivers and microenvironmental interactions would facilitate the treatment of LUAD metastasis in the future.

[1] College of Bioinformatics Science and Technology, Harbin Medical University, Harbin 150086, China. [2] Department of Anatomy, Harbin Medical University, Harbin 150081, China. [3] Basic Medicine College, Harbin Medical University, Harbin 150086, China. [4] Key Laboratory of High Throughput Omics Big Data for Cold Region's Major Diseases in Heilongjiang Province, Harbin, Heilongjiang 150081, China. [5]These authors contributed equally: Xin Li, Lefan Tang. ✉email: qilishuang7@ems.hrbmu.edu.cn; lixia@hrbmu.edu.cn

ung adenocarcinoma (LUAD) is the most common type of lung cancer, with a high mortality rate[1,2]. Metastasis is the main cause of LUAD-related death, and the occult systemic spread of the disease in several patients cannot be detected through routine methods (such as pathological, clinical, and radiological evaluation)[3]. Therefore, a better understanding of the molecular mechanisms underlying LUAD metastasis is imperative for the identification of patients with occult metastases and for the development of customized treatment designs.

Metabolic reprogramming, conducted by cancer cells in response to stress, has been recognized as a requirement for the malignant progression of cancer, and its inhibition has been shown to reduce metastatic spread[4,5]. Recent studies have demonstrated that the metabolic properties and propensities of tumors evolve during cancer progression, allowing tumor cells to acquire cell-autonomous properties associated with enhanced invasiveness or permitting them to alter the microenvironment to accelerate tumor metastasis[6,7], resulting in significant tumor metabolic heterogeneity. In tumor cells, the metabolic switch from oxidative phosphorylation to glycolysis (the Warburg effect) increases their cellular invasiveness and metastasis by promoting epithelial–mesenchymal transition and angiogenesis[8]. Other metabolic adaptations also contribute to the metastatic capabilities of tumor cells[9,10]. Therefore, a more sophisticated view of cancer metastasis-related metabolic reprogramming is required.

The advent of transcriptomic technologies has provided a platform for the exploration of metabolic heterogeneity. A pan-cancer analysis has shown that tumor metabolic heterogeneity is associated with clinical outcomes[11]. A study on pancreatic ductal adenocarcinoma reported a dysregulated glycolysis–cholesterol synthesis axis to be associated with patient outcome[12]. However, cancer metabolism is context-specific[6]. Therefore, whether heterogeneity in distinct metabolic pathways can be used to stratify LUAD into clinically relevant phenotypes, has not yet been determined. Consequently, comprehensive interactions between metabolic phenotypes and the tumor microenvironment are not yet well understood. The emergence of single-cell transcriptomic sequencing (scRNA-seq) has accelerated the discovery of intercellular communications in context-specific diseases, making it possible to systematically elucidate the regulatory mechanisms of metabolic reprogramming in the tumor microenvironment.

This study aimed to identify the metastasis-related metabolic reprogramming phenotypes of LUAD. Genomic analyses were performed to unravel the underlying oncogenic events for the phenotypes. Further analyses of the scRNA-seq data of LUAD samples were used to validate the metabolic evolutionary trajectory from normal to metastatic phenotypes and reveal the intercellular communications.

## Results

### Identifying metastasis-related metabolic reprogramming phenotypes in LUAD.
First, in The Cancer Genome Atlas (TCGA) dataset (Table 1), using Student's t-test with 5% false-discovery rate (FDR), we extracted 431 differentially expressed (DE) genes between the 185 metastatic LUAD samples with lymph node and/or distal metastasis and 209 non-metastatic LUAD samples (Fig. 1a), and found 280 and 151 DE genes to be upregulated and downregulated, respectively. Besides "Cell cycle" and "DNA replication" pathways, the upregulated DE genes also enriched in "Glycolysis/Gluconeogenesis" and "Carbon metabolism" pathways (Hypergeometric distribution model, FDR < 0.05, Fig. 1b). In addition, we found "HIF-1 signaling pathway", associated with glycolysis, to be potentially enriched with the upregulated DE genes (Hypergeometric distribution model, $P < 0.05$, Fig. 1b). In contrast, biological pathways related to lipid metabolism, including "Glycerophospholipid metabolism" and "Ether lipid metabolism", were potentially enriched with the downregulated DE genes (hypergeometric distribution model, $P < 0.05$, Fig. 1c), indicating that lipid metabolism may play different roles from that of glycolysis in LUAD metastasis.

Therefore, we extracted the eight upregulated DE genes (*ALDOA*, *ENO1*, *GAPDH*, *GPI*, *LDHA*, *PGAM1*, *PGM2*, *TPI1*) from the "Glycolysis/Gluconeogenesis" pathway and three downregulated DE genes (*PLPP1*, *GPD1L*, *PLD3*) from the "Glycerophospholipid metabolism" and "Ether lipid metabolism" pathways (see Supplementary Table 1). The 11 DE genes were significantly positively correlated with each other within the same kind of pathway, while significantly negatively correlated with the genes in other kind pathways (Pearson correlation, FDR < 0.05, Supplementary Fig. 1). These results indicated that the imbalance of the glycolysis and lipid metabolism might participate in LUAD metastasis. Therefore, we performed unsupervised hierarchical clustering based on the mRNA expression of the 11 DE genes and found the samples to be clustered into three groups with different glycolytic and lipid metabolic patterns, defined as metabolic reprogramming phenotypes (MPs). The heatmap (Fig. 1d) showed that one MP exhibited lower mRNA expression of glycolytic genes accompanied with higher lipid metabolic genes; one MP displayed higher mRNA expression of glycolytic genes accompanied with lower lipid metabolic genes inversely, and the other MP showed intermediate mRNA expression of all metabolic genes. Whereafter, we calculated the MP-score of each group (see Methods), representing the imbalance of the two kinds of genes' expression, and defined the MP-I–III by scores from the lowest to the highest. Compared to the normal samples, the three phenotypes exhibited gradually increasing mRNA expression of glycolytic genes and decreasing expression of lipid metabolic genes (Fig. 1e).

We found MP-I–III to be significantly enriched in non-metastatic, lymph node metastatic, and distant metastatic samples, respectively (Fisher's exact test, $P < 0.0001$, Fig. 1f). Furthermore, we extracted the stage I treatment-naive patients and found that MP-III had significantly shorter OS than MP-I and MP-II (MP-III vs. MP-I: log-rank $P < 0.0001$; MP-III vs. MP-II: log-rank $P < 0.0001$, Fig. 1g), while there was no significantly different OS between MP-I and MP-II (log-rank $P = 0.4947$, Fig. 1g). Similarly, the percentages of distant metastatic samples were the highest (62.50%) in MP-III and lower in MP-II (27.03%) and in MP-I (13.04%) with a significant difference in the GSE11969[13] dataset (Fisher's exact test, $P = 0.0169$, Supplementary Fig. 2a, b). Whereas the percentages of lymph node metastatic samples were not observed to be significantly different among the three phenotypes (Fisher's exact test, $P > 0.05$).

### Depicting molecular characteristics and oncogenic events for metabolic phenotypes.
Differences in clinical and molecular characteristics (see Methods) across the metabolic phenotypes in the TCGA dataset were evaluated next, and displayed in Fig. 2 whose details are shown in Supplementary Table 2. The percentages of stage IV (15.49%) and stage III (30.99%) in MP-III was obviously higher than that in the other two MPs, the percentage of stage II (33.83%) was the highest in MP-II, whereas the percentage of stage I (58.15%) was the highest in MP-I (Fisher's exact test, $P < 0.0001$). Analogously, the percentages of magnoid subtype, squamoid subtype, and bronchioid subtype (a low invasion subtype[14]) were highest in MP-III (80.28%), MP-II (27.94%), and MP-I (66.31%), respectively (Fisher's exact test, $P < 0.0001$). The hypoxia score[15], as well as stemness[16] and proliferation scores[17], increase progressively from MP-I to MP-III (Wilcoxon rank test, $P < 0.0001$). The immune score[18] was the highest in MP-II, whereas the lowest was in MP-III (Wilcoxon rank test,

**Table 1 The datasets of lung adenocarcinoma used in this study.**

| Data source | Platform | Follow-up information | Sample count |
|---|---|---|---|
| TCGA | Illumina HiSeq 2000 | OS | 394 |
| GSE11969 | Agilent Homo 2.16 K | – | 90 |
| GSE31210 | Affymetrix Plus 2.0 | OS | 162 |
| GSE50081 | Affymetrix Plus 2.0 | OS | 92 |
| GSE13213 | Agilent 4x44K (G4112F) | OS | 79 |
| GSE42127 | Illumina WG-6 V3.0 | OS | 67 |
| GSE68465 | Affymetrix U133A | OS | 223 |
| GSE131907 | Illumina Hiseq 2500 | – | 11 |
| GSE123902 | Illumina HiSeq 2500 | – | 7 |
| CPTAC (mRNA) | Illumina Hiseq 4000 | – | 110 |
| CPTAC (protein) | Tandem mass tags | – | |

*OS overall survival*

$P < 0.0001$). In contrast, the tumor mutation burden (TMB) was observed to be highest in MP-III, while decreased to MP-II and MP-I (Wilcoxon rank test, $P < 0.0001$).

To determine the oncogenic events associated with the different metabolic phenotypes, we further investigated the differences in somatic mutations and copy number variations (CNVs) in the 11 metabolic genes and the 28 known LUAD driver genes (Supplementary Table 3) across the MPs in the TCGA dataset. We found the frequency of mutations in *TP53* and *KEAP1* and CNV deletion in a genomic region containing *SETD2* and *PBRM1* to be significantly different across the three phenotypes (Fisher's exact test, FDR < 0.05, Fig. 2). Of note, all the genetic events occurred more frequently in MP-III than in others.

**Reconfirming the metabolic phenotypes of LUAD**. The above result showed that a percentage of early-stage patients without any metastasis could be clustered into MP-III; thus, we inferred that these patients might have a high risk of metastasis, resulting in poor prognoses. Therefore, we first applied hierarchical clustering on stage I patients derived from five public datasets (Table 1) based on the mRNA expression of the 11 metabolic genes and confirmed the stage I patients also to be clustered into three phenotypes (Fig. 3a–e). Survival analysis (Fig. 3f–i) showed the stage I patients with MP-III to have significantly shorter OS than those with MP-I and/or MP-II in the GSE31210[19] (MP-III vs. MP-I, log-rank $P < 0.0001$, HR = 3.00, 95% CI: 1.56–5.76; MP-III vs. MP-II: log-rank $P = 0.0015$, HR = 6.34, 95% CI: 1.72–23.44), GSE50081[20] (MP-III vs. MP-I, log-rank $P = 0.0174$, HR = 1.86, 95% CI: 1.08–3.20; MP-III vs. MP-II: log-rank $P = 0.0043$, HR = 3.85, 95% CI: 1.42–10.39), GSE13213[21] (MP-III vs. MP-I, log-rank $P = 0.0412$, HR = 1.66, 95% CI: 1.00–2.76; MP-III vs. MP-II: log-rank $P = 0.4347$, HR = 1.58, 95% CI: 0.50–4.97), and GSE42127[22] datasets (MP-III vs. MP-I, log-rank $P = 0.0243$, HR = 2.31, 95% CI = 1.02–5.26; MP-III vs. MP-II: log-rank $P = 0.0754$, HR = 2.64, 95% CI: 0.86–8.10). In the GSE68465[23] dataset, MP-III also exhibited shorter OS but without statistical significance (log-rank $P > 0.05$, Fig. 3j), which might be caused by the lack of one gene (*PGM2*) for clustering in the dataset. Additionally, there was no significant difference in OS between MP-I and MP-II in five independent datasets, but there was a shorter OS tendency of MP-II than that of MP-I in three datasets (GSE13213, GSE42127, and GSE68465). Especially, the GSE13213 dataset (Supplementary Fig. 2c, d) also showed that 30.77% stage I patients in MP-III developed distant metastasis after surgery, marginally significantly higher than that (8.11%) in MP-I (Fisher's exact test, $P = 0.0513$) and tentatively higher than that (22.22%) in MP-II (Fisher's exact test, $P = 0.66$).

Additionally, in a paired proteomic and transcriptomic dataset with 110 LUAD samples, derived from the Clinical Proteomic Tumor Analysis Consortium (CPTAC), we observed the samples were also clustered into the three MPs, based on the protein expression of the 11 metabolic genes (Fig. 4a). Moreover, we found that the mRNA expression of each metabolic gene was significantly positively correlated with its protein expression in the dataset (Pearson correlation, FDR < 0.05, Fig. 4b).

**Displaying transcriptional trajectory of metabolic phenotypes at the single-cell level**. From one scRNA-seq dataset (GSE131907[24]), the 26,436 epithelial cells were cataloged into 14 clusters with 0.1 resolution that the normal cells (Cluster 1, 11, and 13), primary tumor cells (Cluster 4, 8, 9, and 10), and mBrain tumor cells (Cluster 0, 2, 3, 5, 6, 7, and 12) could optimally be separated, which was visualized by UMAP plot (Fig. 5a). Average mRNA expression of the eight glycolytic genes and three lipid metabolic genes for each cell is shown in Fig. 5b, and they were significantly higher and lower, respectively, in mBrain tumor cells than in primary tumor cells and normal cells. Therefore, we subjected the 11 metabolic genes to hierarchical clustering to determine the metabolic phenotypes of the epithelial tumor cells based on MP-score (Fig. 5c, d). The result confirmed the significantly higher percentage of mBrain tumor cells in MP-III than that in MP-I and MP-II (Fisher's exact test, both $P < 0.0001$, Fig. 5e).

We next constructed a metabolic trajectory (Fig. 5f) by ordering the epithelial cells according to the mRNA expression changes in the 11 metabolic genes (see Methods). Seven transcriptional states in the trajectory indicated distinct cellular metabolic fates. The percentages of each MP in the seven states are shown in Fig. 5g. Notably, State 1 and State 4 were both enriched with normal epithelial cells, while the cells in State 4 exhibited very low expression of all the 11 metabolic genes, which was not observed in the bulk dataset (TCGA), thus they were not analyzed in the further analyses. Here, the metabolic trajectory of the glycolysis-lipid imbalance appeared to begin principally from partial normal epithelial cells, which marked the beginning of State 1, evolved to MP-I tumor cells majorly at the end of State 1, and then formed a branched structure with two major cell fates (Cell fate 1 and 2). Tracing the metabolic trajectory of Cell fate 1 (ignoring State 4) revealed that MP-I tumor cells (State 2) evolved into MP-III tumor cells (State 3). Subsequently, following the trajectory of Cell fate 2, we found that along the MP-I tumor cell (State 5) trajectory, MP-II and MP-III tumor cells were mainly located in separate branches of State 6 and State 7 (Fig. 5g). The mRNA expression of glycolytic genes generally increased from pre-branch to State 6 (MP-II) and State 7 (MP-III), while a decrease in lipid metabolic gene expression was only observed from pre-branch to State 7

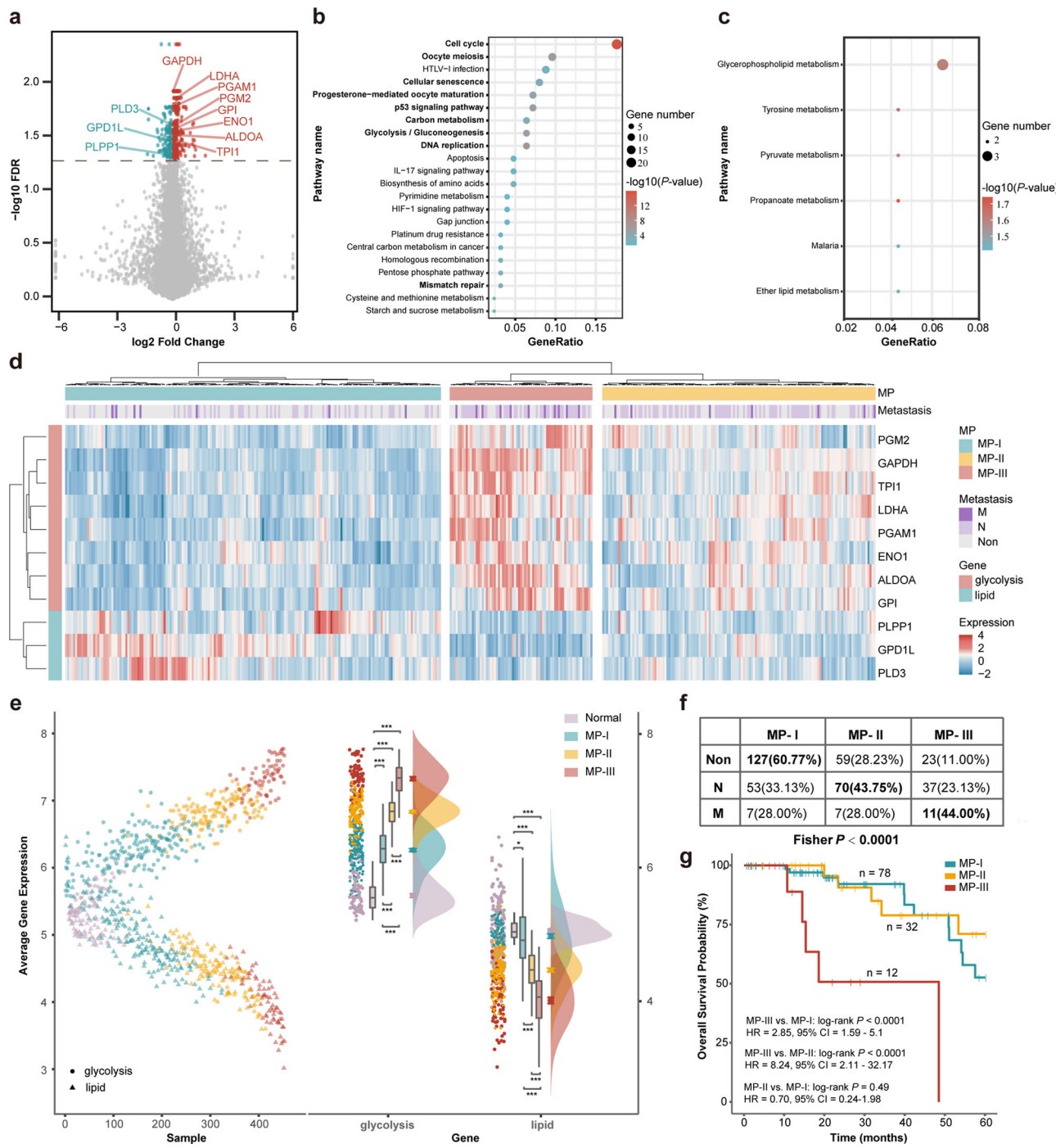

**Fig. 1 Identification of metabolic reprogramming phenotypes (MPs) in the TCGA dataset. a** Volcano plot of differentially expressed (DE) genes between metastatic and non-metastatic samples. Red and blue represented upregulated and downregulated DE genes, respectively. **b, c** Functional pathways enriched with upregulated (**b**) and downregulated (**c**) DE genes. **d** Hierarchical clustering heatmap for all the samples based on the mRNA expression (Z-score) of the 11 DE metabolic genes ($n = 394$ biologically independent samples). Here, the samples were clustered into the three MPs; the one with lower expression of glycolytic genes accompany by higher lipid metabolic genes were defined as MP-I, the one with higher expression of glycolytic genes accompanied with lower lipid metabolic genes was defined as MP-III, and the other one showing intermediate mRNA expression of all metabolic genes was defined as MP-II. **e** Average expression of glycolytic and lipid metabolic genes across the three MPs and normal samples. Boxplots extend from the 25th to 75th percentiles, the line indicates the median, and whiskers indicate the range. **f** Confusion matrix for the number of metastatic samples in the different MPs. **g** Kaplan–Meier curves of overall survival for 122 samples obtained from treatment-naive patients with stage I LUAD.

(Fig. 5h). A similar metabolic trajectory was observed from normal to mBrain (Supplementary Fig. 3).

**Decoding the intercellular communications underlying metabolic phenotypes.** To build a metabolic phenotype-specific

cell–cell communication atlas, we extracted 6372 tumor epithelial cells (MP-I, -II, and -III), 2373 stromal cells (fibroblasts and endothelial cells), and 35,506 immune cells (B lymphocytes, T lymphocytes, NK, myeloid, and MAST cells) from the primary tissues of patients with tLung. In total, we identified 41 significant

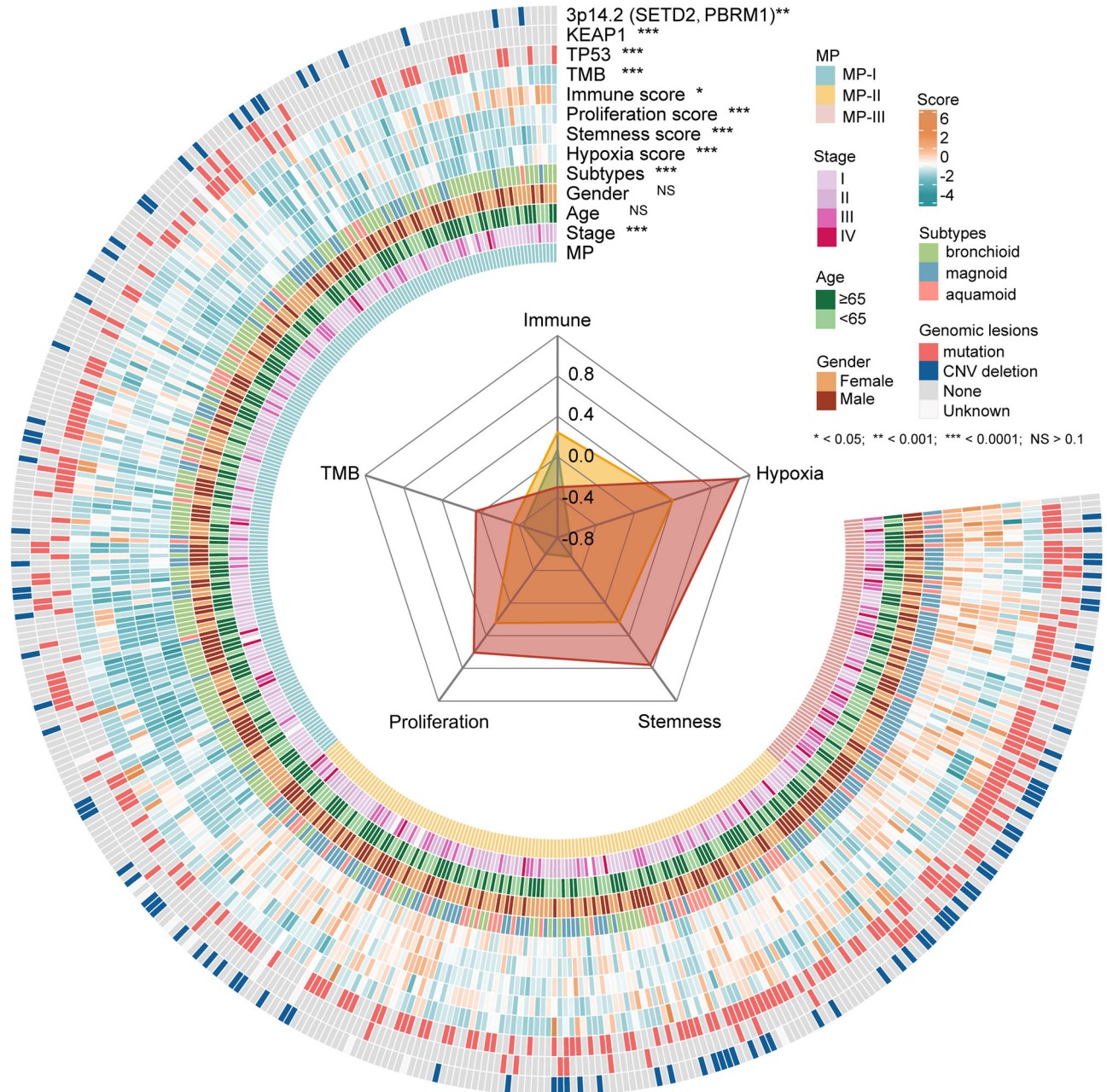

**Fig. 2 Clinical and molecular characteristics of the MPs in the TCGA dataset.** Circle heatmap of clinical information (stage, age, gender, transcriptomics subtypes, molecular scores, and TMB) and oncogenic events (*TP53*, *KEAP1*, *SETD2*, and *PBRM1*). The significance of differences (*P* value) among the three MPs, estimated by Fisher's exact test, is shown after each characteristic. The radar chart (inner) displays four molecular scores and TMB of the three MPs. The detailed numbers of each characteristic in the MPs are presented in Supplementary Table 2.

ligand-receptor pairs between the 10 cell populations (Fig. 6a), and as expected the MP-III population showed the strongest outgoing signal, followed by MP-II, with MP-I weakest (Fig. 6b).

Next, significant ligand–receptor pairs associated with the three metabolic phenotypes were further categorized into 15 signaling pathways (Fig. 6c), including ANGPTL, EGF, SPP1, and VEGF pathways. Network centrality analysis of the inferred ANGPTL signaling network showed the MP-III population to be a unique source of ANGPTL ligands (*ANGPTL4*, *ANGPTL2*) that acted on endothelial cells and fibroblasts (Fig. 7a). The relative contributions of each ligand-receptor pair in the signaling pathway and the role of each cell population are shown in Fig. 7b, c. For the VEGF signaling pathway, all three MP populations were prominent sources of *VEGF* ligands acting on endothelial cells,

and the interaction intensity gradually increased from MP-I to MP-III (Fig. 7d–f). Furthermore, the three MP populations, especially MP-II, were found to be receivers in the EGF signaling pathway, possibly acted upon by themselves as well as by myeloid, MAST, NK, and endothelial cells (Fig. 7g–i). Notably, the MP-III population exhibited significant signal sending activities toward the MP-II population in the EGF signaling.

Collectively, we identified 103 ligand–receptor pairs interacting MP-III with other cell populations (Supplementary Data 1). Similarly, we identified 154 significant ligand–receptor pairs interacting MP-III with other cell populations in an independent scRNA-seq dataset (GSE123902[25]). Thereinto, there were 62 overlapped ligand-receptor pairs, significantly higher than expected (hypergeometric distribution model, *P* < 0.0001, Supplementary

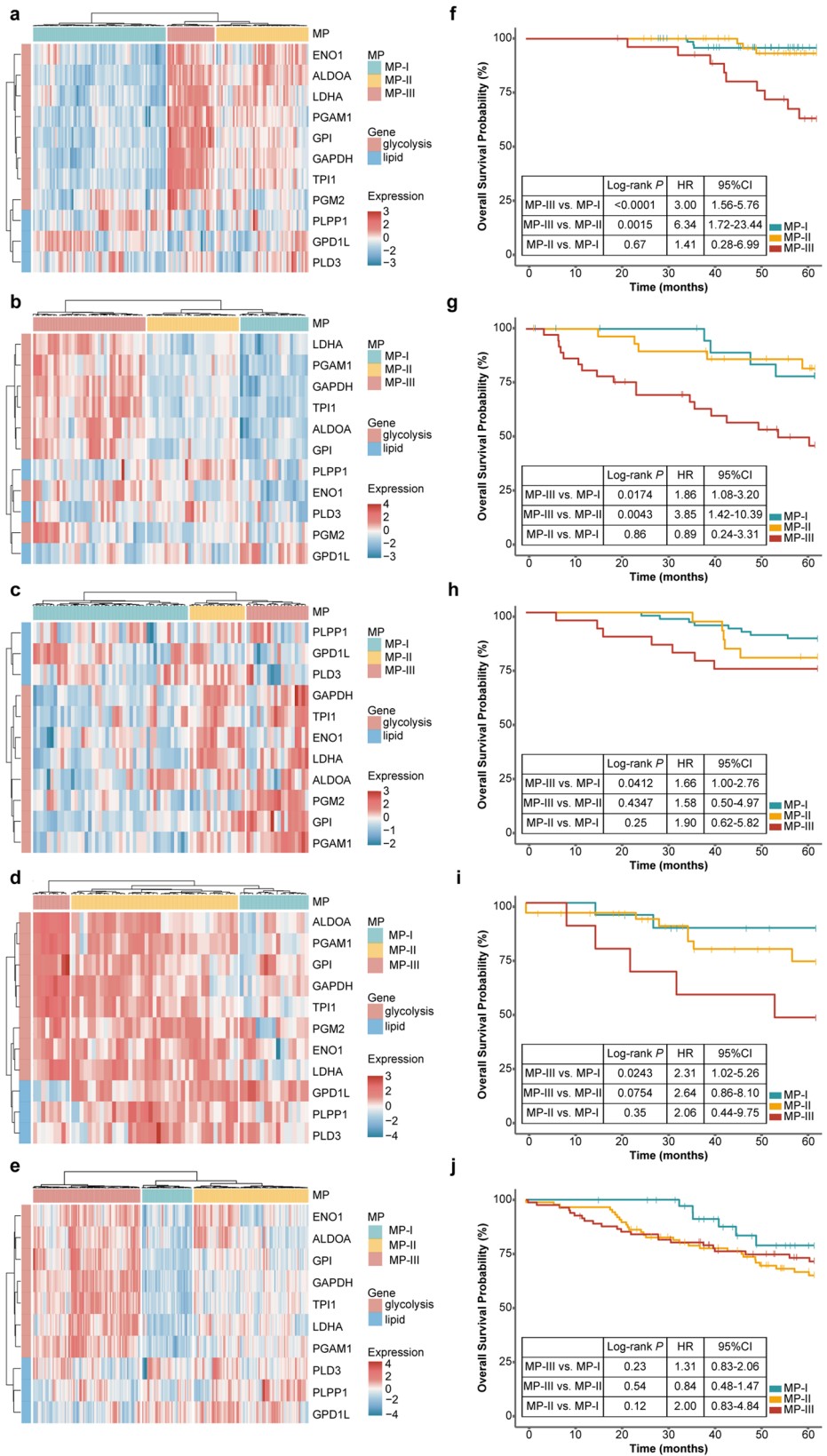

**Fig. 3 Reconfirmation of the MPs and survival in multiple cohorts. a–e** Hierarchical clustering heatmap for treatment-naive patients with stage I LUAD based on the mRNA expression (Z-score) of the 11 metabolic genes in the GSE31210 (*n* = 162 biologically independent samples), GSE50081 (*n* = 92 biologically independent samples), GSE13213 (*n* = 79 biologically independent samples), GSE42127 (*n* = 67 biologically independent samples), and GSE68465 (*n* = 223 biologically independent samples) datasets. **f**, **j** Kaplan–Meier curves of overall survival for the five datasets.

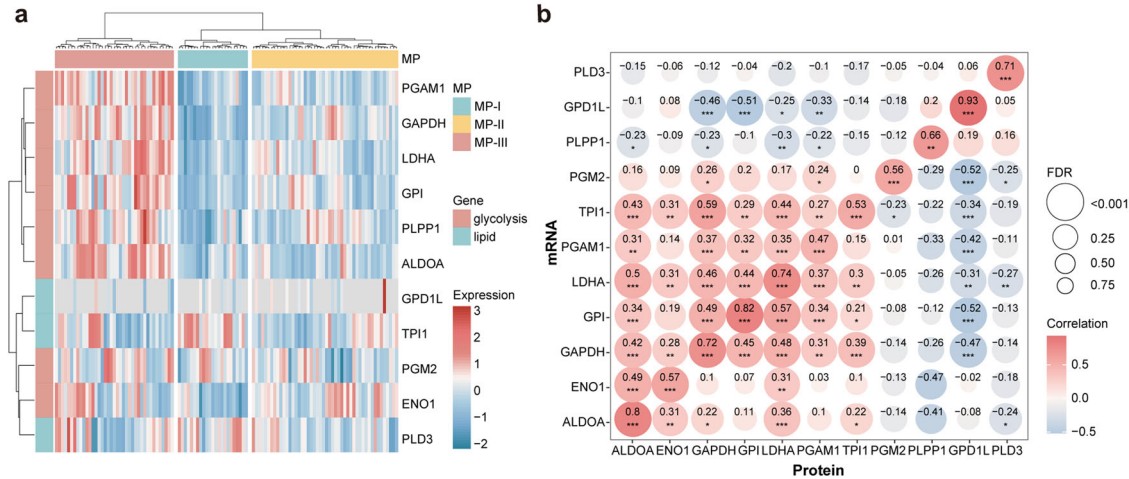

**Fig. 4 Reconfirming of metabolic phenotypes in the proteomic dataset. a** Hierarchical clustering heatmap based on the protein expression (*Z*-score) of the 11 metabolic genes from CPTAC. **b** Correlation matrix for the mRNA and protein expression of the 11 metabolic genes.

Fig. 4). Notably, mRNA expression of nine ligands expressed on MP-III, including *ANGPTL4*, *VEGFA*, etc., was significantly higher than that on MP-I in the GSE131907, which were all validated in another scRNA-seq dataset (Wilcoxon rank test, FDR < 0.05, Supplementary Table 4). Furthermore, in the TCGA dataset, we found the mRNA expression of all the nine ligands to be significantly correlated with the mRNA expression of 11 metabolic genes, which could be regulated by *TP53* and *KEAP1* mutations, and *SETD2* and *PBRM1* deletions (Pearson correlation, FDR < 0.05). A comprehensive network was constructed to link the intracellular oncogenic and metabolic events of MP-III with intercellular communications (Fig. 8).

## Discussion

Metabolic reprogramming is a hallmark of tumor cells and has been recently regarded as a key contributor in tumor progression[26]. In this study, we found that upregulated and downregulated DE genes in metastatic samples were enriched in glycolysis and lipid metabolism pathways, respectively, thereby indicating that the imbalance of glycolysis and lipid metabolism may promote LUAD metastasis. Glycolysis is known to facilitate tumor invasion and metastasis[27]. Among the eight upregulated glycolytic genes, *ALDOA*, *GAPDH*, *LDHA*, and *PGAM1* have been reported to be associated with poor prognosis[28–31]. Lipid metabolism and its metabolites can also induce tumor aggressiveness in LUAD[32]. Therefore, we extracted corresponding eight upregulated glycolytic genes and three downregulated lipid metabolic genes to characterize the metabolic reprogramming patterns of the samples and found LUAD to be clustered into three phenotypes (MPs). MP-III, characterized to have significantly high glycolytic and low lipid metabolic levels, was enriched in distant metastatic samples and associated with the poorest prognosis. Notably, in the TCGA dataset, there was no significant difference in stage I patients' OS between the high- and low-glycolysis groups clustered by eight glycolytic genes (log-rank *P* = 0.66, Supplementary Fig. 5a, b). While, in the high-glycolysis group, the patients with low expression patterns of lipid metabolism exhibited significantly shorter OS than the others (log-rank *P* < 0.0001, Supplementary Fig. 5c, d). These results suggested that high glycolytic and low lipid metabolic levels might synergistically accelerate tumor progression in LUAD. Additionally, we also found that a lot of other glycolytic and lipid metabolic genes were significantly correlated with the MPs (Pearson correlation, FDR < 0.05, Supplementary Data 2), such as a key glycolytic enzyme (*PKM*) gradually increasing from MP-I to

MP-III. Suggesting the stronger dysregulations of the two kinds of pathways in MP-III.

MP-III possessed the highest hypoxia, proliferation, and stemness scores, indicating its high metastatic potency. Notably, MP-III was characterized with the highest TMB but the lowest immune score. Previous studies have found that metabolic dysregulation of tumor cells could suppress the infiltration of immune cells and antitumor immunity[33], resulting in the high TMB tumor translating into an immunologically "cold" tumor[34,35]. Our further genetic analysis revealed that the mutations of *TP53* and *KEAP1*, and deletions of *SETD2* and *PBRM1* might be underlying the metabolic reprogramming. The exploration of the genetic responsibility for MP-III would guide us in designing the studies of a clinical drug combination that may succeed in transforming these patients into those responsive to immunotherapy[35].

The MPs could be confirmed in multiple independent cohorts, and the stage I LUAD patients with MP-III exhibited the worst prognosis and the highest risk of developing distant metastasis after surgery. In contrast, MP-II was inclined to be enriched with lymph node metastatic samples, while it was not observed to have a significantly poorer prognosis than MP-I, which might be explained by the treatment strategy of lymph node dissection after surgery for LUAD[36]. Notably, the independent cohorts were detected with microarray, scRNA-seq, and protein platforms, suggesting the cross-platform robustness of the MPs.

Metastasis imposes metabolic modifications that allow cancer cells to migrate away from primary tissues by altering their microenvironment[37]. Therefore, we first used the scRNA-seq dataset of LUAD to confirm that MP-III was enriched in mBrain tumor cells and to demonstrate a metabolic evolutionary trajectory of epithelial cells from normal to MP-II and MP-III through MP-I. Notably, the evolutionary trajectory exhibited a unique low–low metabolic phenotype (State 4) in the scRNA-seq dataset, which has been reported in melanoma[34], meriting further validation and exploration. Subsequently, the intercellular communication analysis revealed the ANGPTL signaling pathway to be a unique one, in which MP-III population influences endothelial cells and fibroblasts. Padua et al. found that tumor cell-derived *ANGPTL4* disrupted vascular endothelial cell–cell junctions, increased the permeability of lung capillaries, and facilitated the trans-endothelial passage of breast tumor cells[38]. Several studies have also shown the upstream role of *ANGPTL4* with respect to *VEGF*, indicating its role in angiogenesis[39,40]. Our results also showed that all three MP populations could interact with

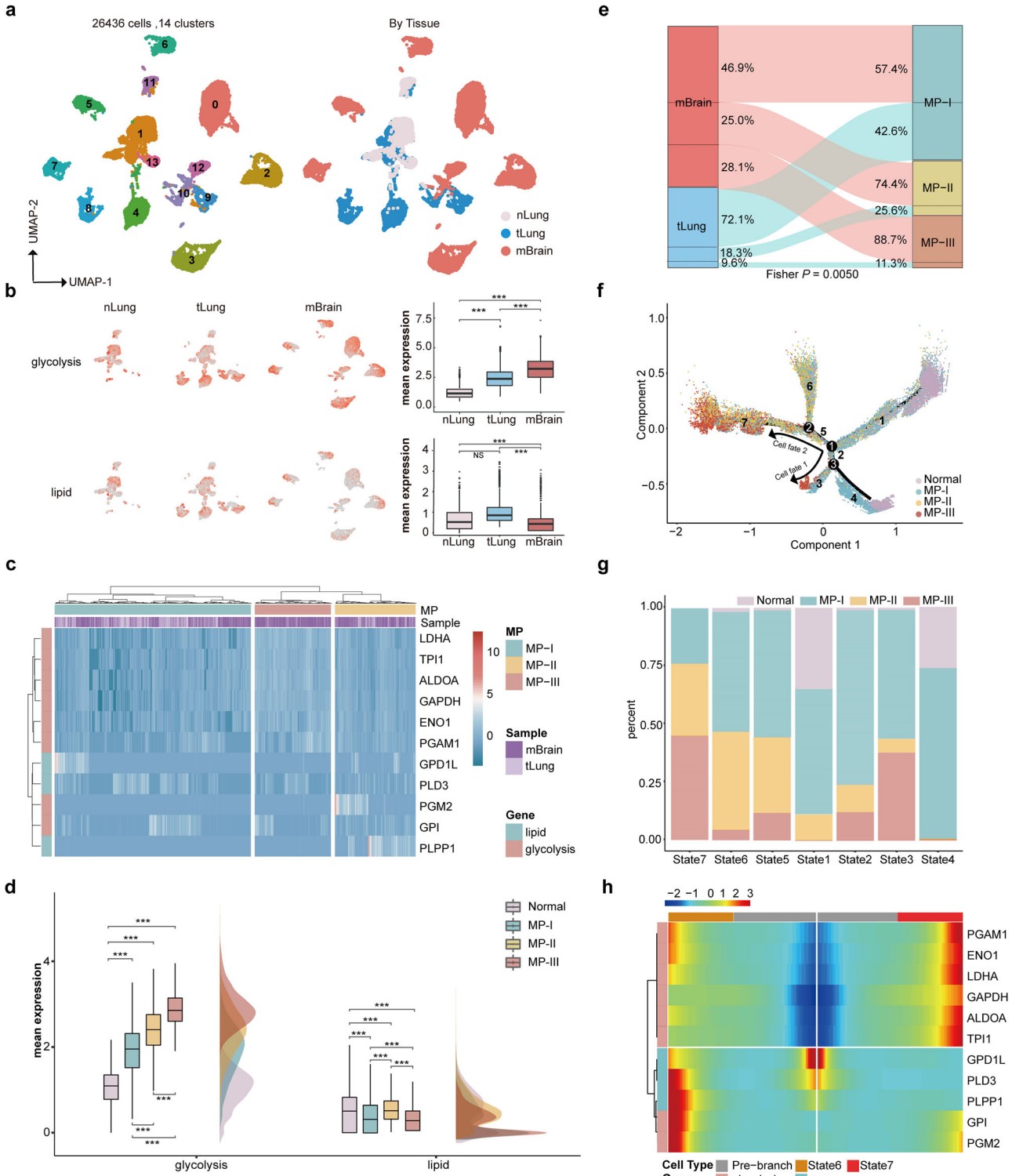

**Fig. 5 Cell trajectory and branched expression analyses for the MPs in the single-cell RNA-sequencing dataset. a** UMAP plot of 26,436 epithelial tumor cells colored based on the major cell lineages and normal, tLung, and mBrain tissues in the GSE131907 dataset. **b** Average expression of the eight glycolytic and three lipid metabolic genes within normal, tLung, and mBrain tissues represented by UMAP plots and box plots. **c** Hierarchical clustering heatmap for epithelial tumor cells based on the mRNA expression (Z-score) of the 11 metabolic genes. **d** Raincloud plot for the average expression of glycolytic and lipid metabolic genes in the three MPs and normal cells. **e** Sankey diagram showing the flow/change of tumor tissues (tLung/mBrain) in the MPs. **f** Unsupervised transcriptional trajectory of epithelial cells based on the 11 metabolic genes using Monocle (version 2), colored based on MP-I, -II, -III, and normal. **g** Relative percentages of the MP populations for each cell state as shown in trajectory. **h** The mRNA expression (Z-score) heatmap for the 11 metabolic genes in a branch-dependent manner. Genes (rows) are clustered and cells (columns) are ordered according to the pseudotime development, which along the pre-branch to State 6 and State 7. Boxplots extend from the 25th to 75th percentiles, the line indicates the median, and whiskers indicate the range.

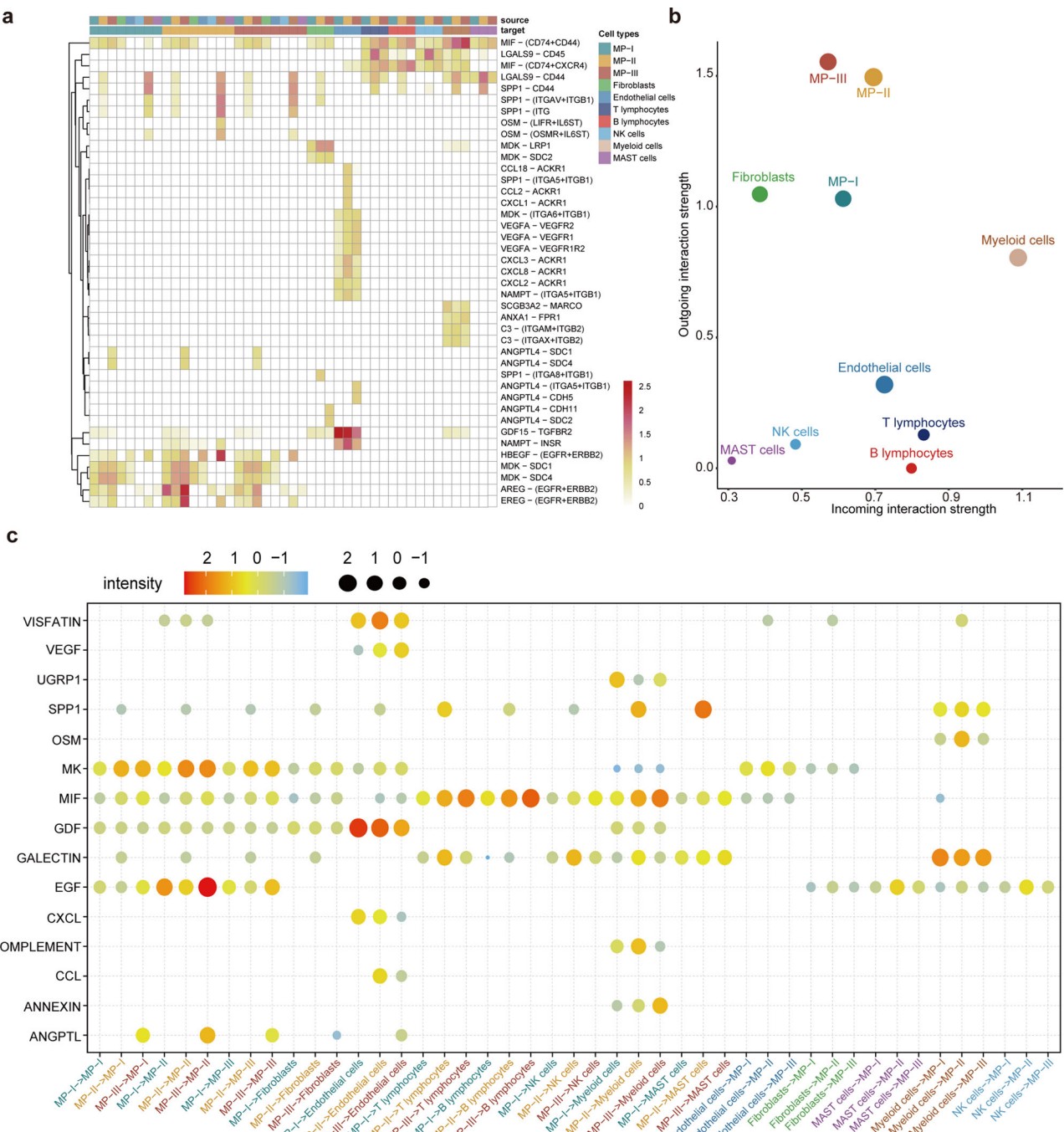

**Fig. 6 Identification of ligand-receptor pairs across 10 cell populations. a** Crosstalk intensity heatmap for significant ligand–receptor pairs across ten cell populations. **b** Dot plot showing the outgoing and incoming signaling patterns of intercellular communications for ten cell populations. **c** Crosstalk intensity heatmap for corresponding signaling pathways of ligand–receptor pairs in each cell population.

endothelial cells in the VEGF signaling pathway and that MP-III exhibited interactions with the most significant intensity. MP-III population not only contributed to its own EGF signaling pathway but also significantly activated the EGF pathway in the MP-II population, indicating the role of MP-III in MP-II deterioration. We also decoded more sophisticated interactome of the MPs with refined subdivided cell subtypes and observed similar results, such as ANGPTL and VEGF signaling pathways (Supplementary Fig. 6, Supplementary Data 3). We additionally observed that Cytotoxic and Exhausted CD8⁺ T lymphocytes be sources of *IFNG* in the IFN-II signaling pathway that acted on MP-II and MP-III populations (Supplementary Fig. 6). Finally, in order to

comprehensively elucidate the underlying molecular alterations for metastasis, we recapitulated a metabolic phenotype-specific cell-cell communication atlas of LUAD and linked it with cell-autonomous oncogenic events (*TP53, KEAP1, SETD2,* and *PBRM1*) through metabolic genes, thereby providing candidate therapeutic targets. Recent studies have shown that mutations in *TP53*[41–43] and *KEAP1*[44] contribute to various metabolic disorders and play pivotal roles in metabolic reprogramming and cancer progression. Furthermore, the protein-protein interaction network from the STRING database (https://string-db.org/) indicated that *TP53* could activate the glycolysis pathway through up-regulating *PGM2*[41], and hereby enhance the intercellular

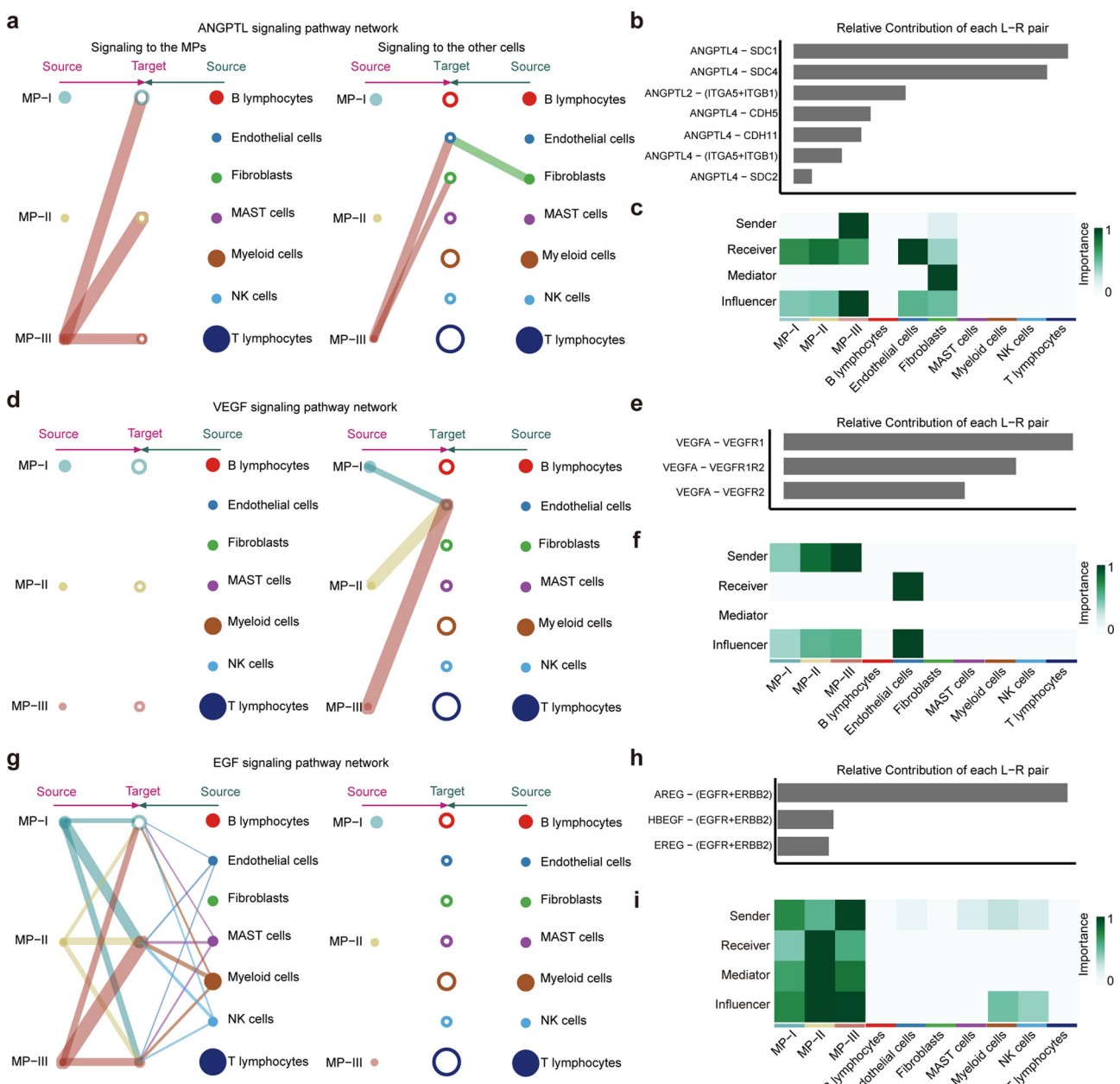

**Fig. 7 Network centrality analysis of specific pathways. a** Hierarchical plot showing the interactions between the MPs and other cells in the ANGPTL signaling pathway. This plot consists of two parts: Left and right portions highlight the autocrine and paracrine to MPs and to other cells, respectively. Solid and open circles represent source and target, respectively. Circle sizes are proportional to the number of cells in each cell group and edge width represents the communication probability. Edge colors are consistent with the signaling source. **b** Relative contribution of each ligand-receptor pair to the overall communication network of the ANGPTL signaling pathway. **c** Relative importance of each cell population based on the computed four network centrality measures of the ANGPTL signaling pathway. Details are described in the original article[57]. **d–f** VEGF signaling pathway. **g–i** EGF signaling pathway.

communication of MP-III with endothelial cells on the VEGF signaling pathway (*VEGFA-KDR*). Meanwhile, *TP53* could regulate lipid metabolism by interacting with *HIPK4* to *PLD3*. Previous studies have shown that the mutation in *KEAP1* could activate *NRF2*, directing epithelial cells toward metabolic reprogramming of glucose metabolism[44,45]. In this study, we found that *KEAP1* mutation might indirectly interact with a glycolytic gene (*LDHA*) through *COPS5*, which is related to the activation of glycolysis[46]. Meanwhile, *KEAP1* might also affect lipid metabolism through indirect interaction with *TP53*. Additionally, the histone H3K36 methyltransferase *SETD2* and the chromatin remodeling factor *PBRM1* are frequently mutated or deleted in a variety of human tumors[47–50]. These studies indicated that

*SETD2* and *PBRM1* deletions might participate in glycolysis-lipid metabolism discoordination by indirectly interacting with *TP53* and *KEAP1*.

This study still had some limitations. First, the metabolic phenotypes defined in this study were based on the mRNA expression of the 11 metabolic genes, thus the activities of glycolysis and lipid metabolisms need to be validated by estimating their metabolites. Second, the cell-cell interactions discovered in this study should be carefully determined, as they were speculative by the mRNA expression of ligands and their receptors. Though the significantly more interactions of the MP-III population with other populations were validated in another scRNA-seq dataset, the relative reliability of the communications still

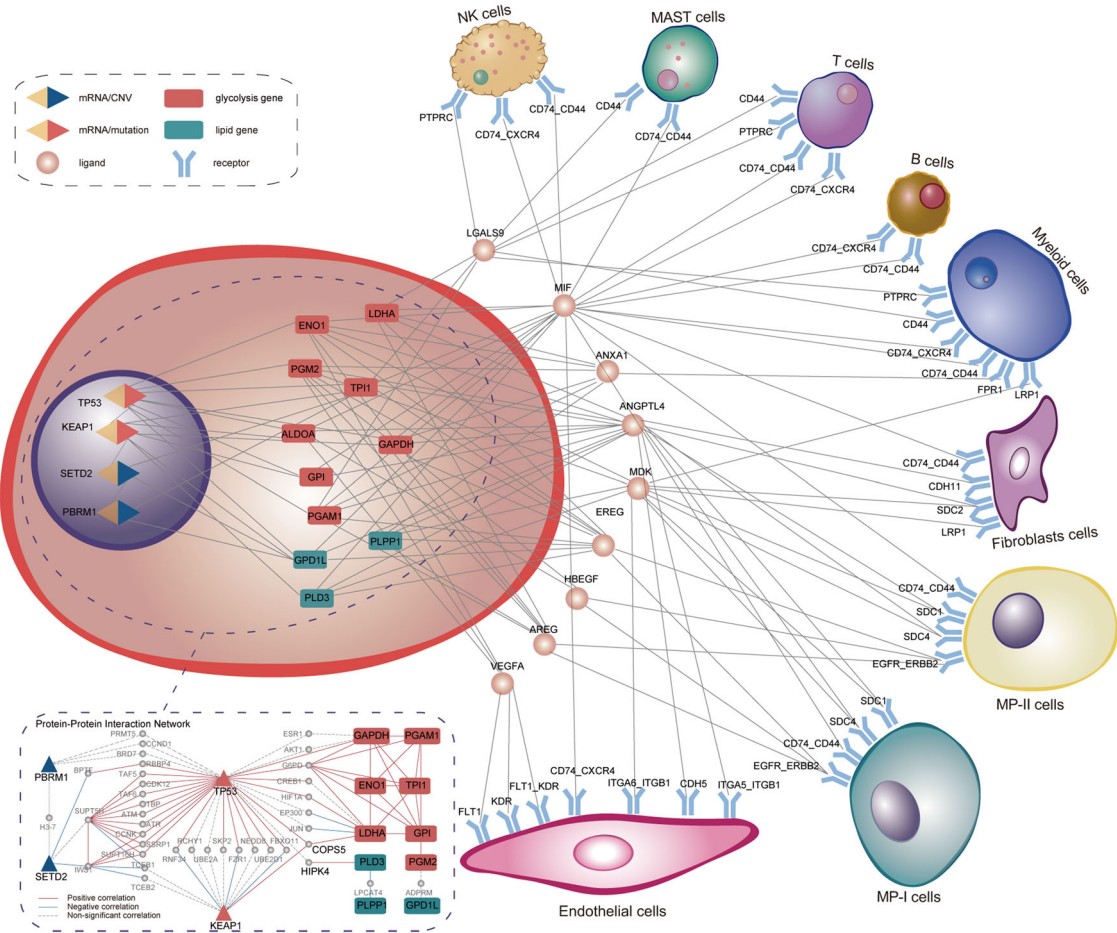

**Fig. 8 Comprehensive network linking the intracellular oncogenic and metabolic events of MP-III with intercellular communications.** The correlation network includes four genetic events (*TP53* mutation, *KEAP1* mutation, *SETD2* deletion, and *PBRM1* deletion), the 11 metabolic genes, and 45 ligand–receptor pairs of MP-III interacting with other cells in the tumor microenvironment. Here, the interacting genes expressed on MP-III were restricted to the nine ligands that were significantly upregulated in MP-III compared to MP-I. The nodes represent these genes and the edges represent the significant correlation between two nodes in the TCGA dataset (Pearson correlation, FDR < 0.05). The dotted box is the protein–protein interaction network between genetic events and the 11 metabolic genes from the STRING database, within the solid lines, represent significant correlation (Pearson correlation, $P < 0.05$).

needs further experimental verification. Furthermore, the other modes of metabolic reprogramming in altering the microenvironment, such as metabolites, merit further exploration.

In conclusion, we determined a metabolic phenotype (MP-III) relevant to the (occult) metastasis of LUAD, which was characterized by glycolytic and lipid metabolic discoordination. The Discovery of the four oncogenic events underlying this metabolic reprogramming and its pivotal interactions (especially the ANGPTL signaling pathway) with other cells provides candidate therapeutic strategies against LUAD metastasis.

## Methods

**Data sources**. In this study, ten LUAD cohorts were obtained from the TCGA (https://portal.gdc.cancer.gov/), Gene Expression Omnibus (GEO, https://www.ncbi.nlm.nih.gov/geo/), and Clinical Proteomic Tumor Analysis Consortium data portal (CPTAC, https://cptac-data-portal.georgetown.edu/). Informed consent was obtained from all participants.

Two metastasis datasets (TCGA and GSE11969) that met the inclusion criteria of including LUAD patients with distant metastases, lymph node metastasis, and non-metastasis, were selected. The TCGA dataset included the mRNA expression profiles of 394 LUAD samples, comprising 25 distant metastatic, 160 lymph node metastatic samples (collectively termed "metastatic samples"), and 209 non-metastatic samples. Among them, mutation and CNV data were available for 390 and 385 samples, respectively. The GSE11969 dataset included the mRNA expression profiles of 18 distant metastatic, 15 lymph node metastatic, and 35 non-metastatic samples. Five other LUAD expression datasets for survival analysis were selected based on the

inclusion criteria that the number of stages I patients not receiving adjuvant therapy after surgical resection (treatment-naive) was more than 50. Therein, the GSE13213 dataset also recorded the follow-up metastasis information of the stage I patients after surgery within five years, among which nine patients developed distant metastases, one patient had lymph node metastasis and 52 patients did not have any metastasis. Clinical information of the patients in the datasets is presented in Supplementary Table 5. Furthermore, we downloaded a paired mRNA and protein expression dataset containing 110 LUAD samples from CPTAC.

In addition, we collected two scRNA-seq LUAD datasets (GSE131907 and GSE123902). In the GSE131907 dataset, we extracted 208,506 cells derived from the primary tissues of 11 patients with tLung, metastatic tissues of 10 patients with mBrain, and 11 distant normal lung tissues. In the GSE123902 dataset, we extracted 18,511 cells derived from the primary tissues of seven patients with tLung.

**Data pre-processing**. For the RNA-seq dataset (TCGA) generated by the Illumina HiSeq 2000 platform, the fragments per kilobase of transcript per million mapped reads values were $\log_2$-scaled plus 1 for gene expression measurements. For the scRNA-seq (GSE131907 and GSE123902) dataset derived from the Illumina HiSeq 2500 platform, the unique molecular identifier count for genes in each cell was log-normalized to transcripts per million-like values, and then $\log_2$-scaled plus 1. Ensembl (ENSG) gene IDs and symbols were mapped to their Entrez gene IDs. For the microarray datasets (GSE31210, GSE50081, and GSE68465) generated by Affymetrix platforms, a robust multi-array average algorithm[51] was used to pre-process the raw data. For the microarray datasets (GSE42127 and GSE13213) generated by the Illumina and Agilent platforms, the originally processed data (series matrix files) were used. Probe IDs were mapped to gene IDs according to the corresponding platform files.

For somatic mutation data derived from the Illumina Genome Analyzer DNA Sequencing GAIIx platform, only 17,821 nonsynonymous mutations were included, and a discrete mutation profile (mutated or not) was generated using Mutect2 somatic variant calls in mutation annotation format. CNV data were processed using the GISTIC algorithm[52], with thresholds of 0.3 and −0.3 for amplified and deleted regions, respectively. CPTAC proteomic data was processed using the Common Data Analysis Pipeline[53].

**Differential expression and functional enrichment analyses**. The student's *t*-test was used to identify the DE genes between two groups. Functional enrichment analysis for DE genes was performed by a hypergeometric distribution model. Functional pathways were downloaded from the Kyoto Encyclopedia of Genes and Genomes signaling pathway database (https://www.genome.jp/kegg/pathway.html).

**Identification of metabolic reprogramming phenotypes**. First, hierarchical clustering was performed on samples to distinguish different clusters following Ward's method, which is based on Pearson correlation between the upregulated glycolytic and downregulated lipid metabolic genes. We pre-deleted the samples (or cells) whose expressions of all DE metabolic genes were zero before clustering and scaled the gene expression. Next, we calculated an MP-score estimating the expression imbalance of the glycolytic and lipid metabolic genes in each cluster as follows:

$$MP - score = \sum_{a=1}^{n}\sum_{i=1}^{k}\frac{G_{ai}}{n*k} - \sum_{b=1}^{m}\sum_{i=1}^{k}\frac{L_{bi}}{m*k} \quad (1)$$

where $G_{ai}$ and $L_{bi}$ represent the expression of *a*th glycolytic gene and *b*th lipid metabolic gene in *i*th sample, respectively; *n* and *m* are the numbers of glycolytic and lipid metabolic genes, respectively, and *k* represents the number of samples in the cluster.

**Correlation and survival analyses**. Wilcoxon rank test was used to determine differences in molecular scores (including hypoxia[15], stemness[16], proliferation[17], and immune scores[18]) and TMB between multiple groups. Here, TMB was estimated as (total number of mutations in genes/total number of bases in genes × $10^6$. Fisher's exact test was performed to evaluate whether the percentages of stages, transcriptomics subtypes[54], and frequencies of genetic lesions were significantly different across the multiple groups. Pearson correlation analysis was used to estimate the correlation of gene expression with another gene expression, gene mutation statuses, CNVs, and protein expression, respectively.

OS was defined as the time from the date of initial surgical resection to the date of death or last contact (censored), which was truncated at 60 months. Survival curves were drawn using the Kaplan–Meier method and were statistically compared using log-rank test[55]. Univariate Cox regression model was used to analyze the association between clinical factors of patients and their OS. HRs and 95% CIs were generated using Cox regression models.

**Single-cell RNA-sequencing data analyses**. The unsupervised clustering of epithelial cells was performed by "RunPCA" and "JackStraw" functions of Seurat package v4.0.1, based on the first 20 principal components of the top 2000 most variable genes among the whole-genome. UMAP ("RunUMAP" function) was used for the visualization of clustering. "Featureplot" function was used to show the average mRNA expression of genes on each cell. The subdivided cell subtypes were identified by SingleR v0.2.0 and manual validated in CellMarker database.

Cell trajectory and branched expression analyses were performed based on the Monocle package (version 2), using the reverse graph embedding machine learning algorithm to learn the changes in 11 metabolic gene expression sequences that each cell must go through as part of a dynamic biological process (here, metabolic reprogramming). The dimensionality of the cells was reduced by the DDRTree method, sequenced in pseudotime trajectory, and finally, visualized[56].

Inference of intercellular communications was analyzed using CellChat package v1.0.0[57]. Ligand–receptor interactions and related signaling pathways were downloaded from the CellChat database (https://www.cellchat.org/). This method inferred the potential interaction intensity of a ligand–receptor pair between two cell populations, which considered their gene expression, signaling cofactors, and cell percentages[57]. Significance was evaluated through a permutation test (1000 times). Only receptors and ligands expressed in more than 25% of cells in the specific cell subsets and those with significantly higher levels in specific cell populations than in other cells, with a threshold of $P < 0.05$ and log2-fold change > 0.5, were analyzed. Outgoing and incoming strengths of each cell population were calculated as the cumulative interaction intensities of ligands and receptors expressed on this kind of cell respectively.

**Statistics and reproducibility**. The details about statistics used in different data analyses performed in this study are given in the respective sections of results and methods. Statistical analyses were performed using R version 3.4.0 (https://www.r-project.org/). *P*-values were adjusted using the Benjamini–Hochberg procedure for multiple testing to control FDR[58]. Statistical significance was defined as two-sided $P < 0.05$ or FDR < 0.05 for multiple testing. Although there was no duplicate

sample, the experimental results were verified in multiple independent datasets, supporting the repeatability of the results.

**Reporting summary**. Further information on research design is available in the Nature Research Reporting Summary linked to this article.

## Data availability
All datasets generated or analyzed during this study were collected from public databases, shown in Table 1 and Supplementary Table 5, which could be downloaded from the official website of the databases (see Methods). Data underlying figures in the main text are presented in Supplementary Data 4.

## Code availability
Code utilized in these analyses is immediately available from the corresponding author upon reasonable request.

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

## Acknowledgements

This study was supported by grants from the National Natural Science Foundation of China (81872396 to L.Q., 61873075, 32070673 to X.L.); the National Key R&D Program of China (2018YFC2000100); the Postdoctoral Scientific Research Developmental Fund (grant number LBH-Q16166 to Y.G.); Heilongjiang Touyan Innovation Team Program; National Undergraduate Innovation and Entrepreneurship Training Program (202110226018 to L.Q.).

## Author contributions

X.L. (Xia Li) conceived the idea, L.S.Q. designed the experiments and modified the paper, X.L. (Xin Li), L.F.T designed and performed the experiments, wrote the paper, J.X.D., X.Y.Q. and J.X.Z. analyzed the gene expression data, H.T.Q., M.Y.L. and Y.X.L. pre-processed the radiogenomic data, W.Y.Z. and Y.Y.G. understanding the results. All authors approved the final version.

## Competing interests

The authors declare no competing interests.
