## [Peer Review File · Communications Biology]

Reviewers' comments:

Reviewer #1 (Remarks to the Author):

Comments to Authors

The manuscript by Dr. Lishuang Qi presents with several relevant issues that I will highlight below, in a per-point manner:

1. First of all, all Figures are not thoroughly described in Results, and Legends are missing. The authors cherry-pick a few items they want to describe per Figure but they do not explain all the data presented in each Figure. In detail, Figures 1-2 is poorly described; In Fig2 MP1-3 display a different colors that the one presented in the legends; Figure 3 does not appear to add anything new, as compared to others; Figure 4I is difficult to comprehend.
2. Their conclusions are drawn by mixed dataset: RNAseq and Microarrays. They should at least describe that, and explain what is gathered by individual dataset.
3. It does not make sense and the authors need to explain why there's change in number of clusters used and clearly justify the change, in tSNE plots and trajectory. They should also explain why they adopted tSNE plots, how they came to number of clusters, since with tSNE they can be arbitrary, so we need a justification why they settled on 13 clusters, what is the rationale? Also, why tSNE? People are now typically using umap, that should be justified as it is no more commonly seen.
4. Not clear why they specifically chose 8 upregulated genes involved in glycolysis, and 3 downregulated genes involved in lipid metabolism to characterize the metabolic reprogramming. Their subdivision in MP1-2-3 is arbitrary. The 11 genes should be named in the text, and their relevance should be explained. Why these specific 11 genes were selected?
5. MP1-3 should be defined, applying enricher or IPA, what are they exactly? What happens to other glycolysis or lipid genes in those populations?
6. Speculate why in Figure 1G MP1 has worse survival than MP2.
7. It is unclear why (lines 156-159) MP1, 2 and 3 contain similar met samples and recurrent samples. First of all, they should explain what do they mean by "recurrent". Subsequently they should explain how the above makes sense with MP3 being more "severe" than MP1.
8. Explain why they selected GSE11969 for validation (line 154)? Why only such dataset? If other dataset are utilized, how reproducible are their data?
9. In describing Figure 2 they mention MP1 and MP3 data, what about MP2? Everything that is presented in Figures must be described and discussed.
10. MP3 is associated to TP53, KEAP and SETD2 anomalies. Why do they talk about "4 genes"? They say "MP3 have lesions in at least one of the 4 genes". TMB is not a gene. In addition, what is the relevance that 87.3% of MP3 have at least one mutation in those genes?
11. Table S2 require % not only numbers. Stage IV has equal distribution in MP1 and MP2 and it is just slightly higher in MP3; this is not convincing. Only stage 2 is higher in MP2 (versus what? Significance versus MP1 or MP3? It really is not clear). They mention in text that MP3 is enriched in squamoid (not aquamoid, as written) and magnoid. Actually, MP2 is enriched in those subgroups, according to the figure. It is all presented in a very confusing and not carefully organized manner. Also, bronchioid is not enriched in MP1 as stated in text, but it is in MP3, if I am interpreting correctly suppl table2. They should explain differences in TMB and immune score, it is unclear why the first parameter is high and the second low in MP3.
12. Explain why the 2 other dataset (I and J) in Figure 3 do not show differences in OS. Which differences are there in these 2 datasets that make up for that observation? It is rather odd that MP3 stands out in F and G, that according to K and L, only contain stage I and II tumors. My understanding is that MP3 should contain more aggressive stages....
13. Also, what is the value/relevance of classifying tumors in MP subpopulations with different levels of aggressiveness, if we need staging to get differences in OS?
14. Figure 4D: not clear how the level of significance can be so high in the lipid panel
15. I assume by gene expression they mean RNA expression (line 216)?
16. What does the PET radiomic dataset add?
17. Explain SUVmax, MTV, and TLG levels when mentioning them
18. Clusters are not described well: cluster 1 is a mix of normal and tumor; Normal is also contained in a small subcluster of cluster 5
19. The lipidic genes are the same in normal and brain mets... really a signature of 3 genes is not so realistic to adopt

20. Figure 4c: here the branching describing the the clustering is different than in previous figures. Here MP2 and MP3 are coming out from the same branch and MP1 is separate. Instead in the previous figures, MP1 clustered closer to MP2, and MP3 was separate. How to interpret that?
21. Rather than association with MP3 they observed association with the glycolysis genes.
22. Interpretation of Trajectory is not convincing. How can you find normal cells at the tip of a trajectory adopted to explain how tumors evolve? If they suspect there was noise during single cell sequencing they should disregard all the data. How do they exclude their other data are not "contaminated" by noise? Description of Fig 4G is extremely poor.
23. Fig 4I: same colors for state 7 and glycolysis
24. Interactome: the immune cells can be subdivided further, plus what is the lymphocyte activation state? The TME contribution may contain less active immune cells, also depending from tumor stage....
25. Interactome Fig 5b: describe and speculate better on the data. They say almost nothing about all the data, they just cannot say that MP2 and MP3 talk more...what are the different contribution of all the subpopulation they show? biological relevance or at least hypothesis on the observations?? The entire interactome is just speculative. As presented it is not a useful information.
26. Stratification of interactions with known mutation in epithelial cells?

Reviewer #2 (Remarks to the Author):

In general I think this is good paper which focuses needed attention on the specific role of metabolic phenotypes in differentially enabling metastasis. Most of the results are reasonably convincing and hence I only have some technical requests/comments

- Can the authors be more specific as the criteria used to pick the 11 genes used for phenotype classification, from the larger set of differentially expressed genes.

-Could the authors provide some more information about the method used to infer transcriptional trajectories (p. 14). The results of this section I found to be less convincing than other aspects of the paper. For example, is there any way to distinguish gene dropouts (line 258) from the possibility the some cells exhibit a low metabolic phenotype (as seen in other situation, for example Jia, Dongya, et al. "Drug-tolerant idling melanoma cells exhibit theory-predicted metabolic low-low phenotype." *Frontiers in oncology* 10 (2020): 1426.). And, is there an overall picture that one can be certain of, given the murkiness of the findings in Fig 4

-I think it would be useful to present some more discussion of the role of the specific genetic modifications as being partially responsible for the metabolic changes. For example, KEAP is known to be an inhibitor of NRF2 and hence can control the level of ROS in cells; TP53 can affect the EMT status of cells and its malfunction can help drive a more glycolytic state and more metastatic competence

-Do the authors have any ideas as to how to directly verify the results using CELLCHAT?
I am somewhat skeptical that ligand-receptor expression levels are a reliable indicator of actual information transfer between cells and I would have liked some level of validation. Also, there are clearly other modes cell-cell communication, for example through metabolites, which cannot be captured by expression levels, A more complete discussion would be helpful here.

Reviewer #3 (Remarks to the Author):

In this manuscript, Xin Li et al. conducted transcriptome analysis from lung adenocarcinoma and identified 11 genes which were related in three metabolic reprogramming phenotypes and associated with metastasis. They successfully classified patients in TCGA and independent public datasets with transcriptome or proteome information. They also performed analysis of one scRNA-seq dataset and identified intercellular interaction which was related in metabolic phenotypes MP-I, MP-II and MP-III. There are some points that need to be addressed to improve the manuscript as

below;

Points;

1. The authors extracted 11 DE metabolic genes from a lot of DE genes. I think that this step is one of the most critical steps in this study. Please describe the details how to select eight glycolysis and three lipid metabolism genes. Are there only 11 genes associated with these metabolic pathways in all DE genes?

2. The authors identified metabolic phenotypes and their commutations with stromal cells and immune cells using scRNA-seq data. However, details of metabolic pathways and these changes in the three phenotypes are still unknown. The authors additionally need to describe patterns of pathways including genes and metabolites associated with the 11 metabolic genes and the three phenotypes in lung adenocarcinoma.

3. I recommend that they would add the discussion about master regulators of the changes of the metabolic phenotypes and occurrence of metastasis.

4. The authors found that KEAP1 mutation statuses were associated with the metabolic phenotypes. I think that KEAP1 is associated with oxidative stress responses and mutations of this gene tend to affect several metabolic pathways. Is there any direct association between KEAP1 mutations and the detected metabolic changes of glycolysis and lipid metabolism?

5. For intercellular communication, the authors validated expression patterns of 17 genes only using bulk transcriptome datasets of TCGA data. The detected ligand-receptor interaction should be validated using different scRNA-seq datasets of lung adenocarcinoma or by other experimental methods.

6. I could not find figure legends.

Minor points;

1. Line 113, p. 7: with with -> with

2.

3. Line 181, pp. 11: I could not find the list of 28 LUAD driver genes.

4. Line 463, pp. 24: the description for pre-processing of gene mutation data was insufficient.

Reviewers' comments:

Reviewer #1 (Remarks to the Author):

Comments to Authors

The manuscript by Dr. Lishuang Qi presents with several relevant issues that I will highlight below, in a per-point manner:

1. First of all, all Figures are not thoroughly described in Results, and Legends are missing. The authors cherry-pick a few items they want to describe per Figure but they do not explain all the data presented in each Figure. In detail, Figures 1-2 is poorly described; In Fig2 MP1-3 display different colors that the one presented in the legends; Figure 3 does not appear to add anything new, as compared to others; Figure 4I is difficult to comprehend.

Reply: Thank you for these insightful comments and suggestions. We are sorry that the figure legends were not reviewed with the text, which brings greatly confusion for understanding the figures. We have added the figure legends at the end of the revised manuscript. In accordance with your suggestion, we have described all the results represented in each Figure in the revised manuscript.

The legend of Figure 1 was as follows on page 35: “**Figure 1. Identification of metabolic reprogramming phenotypes (MPs) in the TCGA dataset. (A) Volcano plot of differentially expressed (DE) genes between metastatic and non-metastatic samples. (B-C) Functional pathways enriched with upregulated (B) and downregulated (C) DE genes. (D) Hierarchical clustering heatmap for the 394 samples based on the mRNA expression of the 11 DE metabolic genes. Here, the samples were clustered into the three MPs; the one with lower expression of glycolytic genes accompany with higher of lipid metabolic genes was defined as MP-I, the one with higher expression of glycolytic genes accompany with lower of lipid metabolic genes was defined as MP-III, and the other one showing intermediate mRNA expression of all metabolic genes was defined as MP-II. (E) Average expression of glycolytic and lipid metabolic genes across the three MPs and normal samples. (F) Confusion matrix for the number of metastatic samples in the different MPs. (G) Kaplan–Meier curves of overall survival for 122 samples obtained from treatment-naive patients with stage I LUAD.**” We have described all the results presented in the figure in the revised Results section, for example, we described Fig. 1D in detail on page 6: “**Therefore, we performed unsupervised hierarchical clustering based on the mRNA expression of the 11 DE genes, and found the samples to be clustered into three groups with different glycolytic and lipid metabolic patterns, defined as metabolic reprogramming phenotypes (MPs). The heatmap (Fig. 1D) showed that one MP exhibited lower mRNA expression of glycolytic genes accompany with**

higher of lipid metabolic genes; one MP displayed higher mRNA expression of glycolytic genes accompany with lower of lipid metabolic genes inversely; and the other MP showed intermediate mRNA expression of all metabolic genes.”

The legend of Figure 2 was as follows on page 36: “Figure 2. Circle heatmap of clinical information (Stage, Age, Gender, Transcriptomics subtypes, Molecular scores, and TMB) and oncogenic events (*TP53*, *KEAPI*, and *SETD2*). The significance of differences (*P* value) among the three MPs, estimated by Fisher’s exact test, is shown after each characteristic. Radar chart (inner) displays four molecular scores and TMB of the three MPs.” We are sorry for the mistake in Figure 2, and we have revised the colors of MPs in revised Figure 2.

Figure 3 represents the differences in overall survival between the three phenotypes in the patients without any metastasis (N_0M_0) across multiple datasets, in order to support the inference that MP-III with high-risk of distant metastasis had significantly poor prognosis than the other two phenotypes. We have described the inference in the revised Results section on page 9: “The above result showed that a certain percentage of early-stage patients without any metastasis could be clustered into MP-III; thus, we inferred that these patients might have high-risk of metastasis during the development, resulting in poor prognoses.”

Figure 4I exhibited the mRNA expression of glycolytic genes generally increased from pre-branch (State 1 and State 5) to State 6 (left) and State 7 (right), while a decrease in lipid metabolic gene expression was only observed to State 7. We have described it in the revised Results on page 12: “The gene expression of the 11 metabolic genes on the cells changing from pre-branch to State 6 and State 7, respectively, are displayed in Figure 5H; the mRNA expression of glycolytic genes generally increased from pre-branch to State 6 (MP-II) and State 7 (MP-III), while a decrease in lipid metabolic gene expression was only observed from pre-branch to State 7 (MP-III).” and in the legend of Figure 5 on page 35: “(H) The mRNA expression heatmap for the 11 DE metabolic genes in a branch-dependent manner. Genes (rows) are clustered and cells (columns) are ordered according to the pseudotime development, which along the pre-branch to State 6 (on the left) and State 7 (on the right).”

2. Their conclusions are drawn by mixed dataset: RNAseq and Microarrays. They should at least describe that, and explain what is gathered by individual dataset.

Reply: We thank the reviewer for this suggestion. In this study, we gathered datasets with whole-genome expression profiles detected by RNA-sequencing or microarray from TCGA and GEO databases. We defined the metabolic reprogramming phenotypes in a RNA-sequencing dataset with the largest sample size (TCGA) and validated the phenotypes in five independent microarray datasets and a single-cell transcriptomic

sequencing dataset. This result indicated the robustness of the metabolic reprogramming phenotypes in different datasets assessed with different platforms. According to the reviewer's suggestion, we have discussed this issue in the revised discussion on page 17: **“Notably, the independent cohorts were detected with microarray platforms, different from that detected in the discovery dataset (RNA-seq), suggesting the cross-platform robustness of the MPs.”** Additionally, we have described the inclusion criteria of gene expression datasets used for survival analysis in the revised Methods section on page 21: **“Five other LUAD expression datasets (GSE31210, GSE50081, GSE13213, GSE42127, and GSE68465) for survival analysis, were selected based on the inclusion criteria that the number of stage I patients not receiving adjuvant therapy after surgical resection (treatment-naive) was more than 50.”**

3. It does not make sense and the authors need to explain why there's change in number of clusters used and clearly justify the change, in tSNE plots and trajectory. They should also explain why they adopted tSNE plots, how they came to number of clusters, since with tSNE they can be arbitrary, so we need a justification why they settled on 13 clusters, what is the rationale? Also, why tSNE? People are now typically using umap, that should be justified as it is no more commonly seen.

Reply: We are sorry for the unclear descriptions of tSNE plots and trajectory. In the scRNA-seq dataset, we used different clustering methods for the epithelial cells based on different gene sets to present the specific results, which were displayed in the tSNE plots and trajectory, respectively. Therefore, the numbers of clusters were different since the clusters are highly dependent on the genes used for the unsupervised clustering.

First, we performed the unsupervised clustering of epithelial cells based on the first 20 principal components, obtained from the top 2000 most variable genes using RunPCA and JackStraw functions of Seurat package v4.0.1, and found that the epithelial cells were optimally clustered into 14 clusters (cluster 0-13) that the cells from normal, primary and brain metastasis tissues could be optimally separated from each other and gather internally with the resolution = 0.1. Then, the tSNE plots were used to visualize this clustering result of different kinds of epithelial cells, and further display the gene expression intensities of glycolytic and lipid metabolic genes on these cells to support the result that the expression of these genes was associated with metastasis. According to the reviewer's suggestion, we have adopted the UMAP method for the visualization of clustering result in the revised Figure 5. According to the reviewer's suggestion, we described the method for clustering more clearly in the revised Methods section on page 25: **“The unsupervised clustering of epithelial cells was performed by RunPCA and JackStraw functions of Seurat package v4.0.1, based on the first 20 principal components of the top 2000 most variable genes among the whole-genome. The resolution was set at 0.1, thus epithelial cells derived from normal, primary tumor and**

mBrain tumors were optimally separated, yielding a total of 14 clusters (cluster 0-13). UMAP was used for the visualization with RunUMAP function of Seurat. Featureplot function of Seurat was used to show the average expression of glycolytic genes and lipid metabolic genes on each cell, respectively.”

Different from the clustering result visualized by UMAP, we employed the reverse graph embedding machine learning algorithm of Monocle package to learn a metabolic pseudotime trajectory of epithelial cells, which determined distinct cell state based on the changes in gene expression levels of the 11 metabolic genes. We have described these more clearly in the revised Method section on page 25: “Cell trajectory and branched expression analyses were performed based on Monocle (version 2), using the reverse graph embedding machine learning algorithm to learn the changes in 11 metabolic gene expression sequences that each cell must go through as part of a dynamic biological process (here, metabolic reprogramming). The dimensionality of the cells was reduced by the DDRTree method, sequenced in pseudotime trajectory, and finally visualized [52].”

4. Not clear why they specifically chose 8 upregulated genes involved in glycolysis, and 3 downregulated genes involved in lipid metabolism to characterize the metabolic reprogramming. Their subdivision in MP1-2-3 is arbitrary. The 11 genes should be named in the text, and their relevance should be explained. Why these specific 11 genes were selected?

Reply: We apologize for the unclear description about the selection of the 11 genes. In this study, we found that the upregulated DE genes were significantly enriched in biological pathway related to glycolysis, while the downregulated DE genes were potentially enriched in lipid metabolism, suggesting the imbalance of glycolysis and lipid metabolism during metastasis. Previous studies showed the discoordination of glycolysis and lipid metabolism of tumors to be associated with clinical outcome. Therefore, we focused on the upregulated DE genes in the glycolysis and downregulated DE genes in the two enriched lipid metabolism pathways (‘Glycerophospholipid metabolism’ and ‘Ether lipid metabolism’), to explore the metabolic reprogramming phenotypes. According to the reviewer’s suggestion, we have described this process more clearly and named the 11 genes in the revised Results section on page 5: “Previous studies showed the discoordination of glycolysis and lipid metabolism of tumors to be associated with clinical outcomes [13]. Therefore, in this study, we focused on the upregulated genes involved in glycolysis and downregulated genes in lipid metabolism to analyze the metabolic reprogramming associated with LUAD metastases. Totally, we extracted the eight upregulated DE genes (*ALDOA*, *ENO1*, *GAPDH*, *GPI*, *LDHA*, *PGAM1*, *PGM2*, *TPPII*) from the ‘Glycolysis/Gluconeogenesis’ pathway and three downregulated DE genes (*PLPP1*, *GPDIL*, *PLD3*) from the two lipid metabolism pathways (‘Glycerophospholipid metabolism’ and ‘Ether lipid metabolism’). Here, the detailed functions of the 11 DE genes are represented in Table S1. Additionally, we analyzed the

mRNA expression correlation of these genes and found that the 11 DE genes were significantly positively correlated with each other within the same kind pathway, while significantly negatively correlated with the genes in other kind pathways (Pearson correlation, FDR < 0.05, Fig. S1). These results indicated that the imbalance of the glycolysis and lipid metabolism might participate in LUAD metastasis.”

Additionally, we have explored the biological and expression relevance of the 11 DE genes, and displayed in the revised Table S1 and Figure S1, respectively.

We have rephrased the definition of MPs more clearly in the revised Results section, which have been described in the response to the comment 5.

5. MP1-3 should be defined, applying enricher or IPA, what are they exactly? What happens to other glycolysis or lipid genes in those populations?

Reply: We apologize for the unclear definition of MPs. The MPs were defined by the differences of average gene expression between glycolytic and lipid metabolic genes in each group, where the group with the minimum difference was defined as MP-I, the group with the maximal difference was defined as MP-III, and the other intermediate group was defined as MP-II. In order to described more clearly, we have proposed a metabolic reprogramming score for the definition of MPs in the revised Methods section on page 24: “Next, we calculated a metabolic reprogramming score (MP-score) estimating the expression imbalance of the glycolytic and lipid metabolic genes in each cluster as follows:

$$MP - score = \sum_{a=1}^n \sum_{i=1}^k \frac{G_{ai}}{n * k} - \sum_{b=1}^m \sum_{i=1}^k \frac{L_{bi}}{m * k}$$

where G_{ai} and L_{bi} represent the expression of a th glycolytic gene and b th lipid metabolic gene in i th sample, respectively; n and m are the numbers of glycolytic and lipid metabolic genes, respectively, and k represents the number of samples in the cluster.”

We also rephrased the description in the revised Result section on page 6: “Therefore, we performed unsupervised hierarchical clustering based on the mRNA expression of the 11 DE genes, and found the samples to be clustered into three groups with different glycolytic and lipid metabolic patterns, defined as metabolic reprogramming phenotypes (MPs). The heatmap (Fig. 1D) showed that one MP exhibited lower mRNA expression of glycolytic genes accompany with higher of lipid metabolic genes; one MP displayed higher mRNA expression of glycolytic genes accompany with lower of lipid metabolic genes inversely; and the other MP showed intermediate mRNA expression of all metabolic genes. Whereafter, we calculated the MP-score of each group (see Methods), representing the imbalance of the two kind genes’ expression, and defined the MP with

the lowest score as MP-I, that with the highest score as MP-III, and the remaining MP was defined as MP-II.”

In accordance with your suggestion, we analyzed other glycolytic and lipid metabolic genes in corresponding biological pathways among three MPs, and found that 26 glycolytic genes were significantly positively correlated with MPs that gradually increased from MP-I to MP-III through MP-II (Pearson correlation, FDR < 0.05), significantly higher than the number of the negatively correlated genes (Binomial test, $P = 0.0266$), suggesting the activation state of glycolysis. Notably, pyruvate kinase M1/2 (PKM), a key glycolytic enzyme, was gradually increased from MP-I to MP-III. Similarly, 43 lipid metabolic genes were significantly negatively correlated with MPs, significantly higher than the number of positively correlated genes in the two lipid metabolic pathways (Binomial test, $P = 0.0026$). The result showed that there were more dysregulated genes involved in the two kinds of pathways between the three MPs, suggesting the stronger dysregulations of the two kinds of pathways in MP-III. These results have been mentioned in the revised Discussions on page 16: “Additionally, we also found that a lot of other glycolytic and lipid metabolic genes in corresponding biological pathways were significantly positively and negatively correlated with MPs (Pearson correlation, FDR < 0.05, Table S6), such as a key glycolytic enzyme (PKM) gradually increasing from MP-I to MP-III. The results indicated the stronger dysregulations of the two kinds of pathways in MP-III. However, whether more glycolytic or lipid metabolic genes could be used to improve the definition of MPs needs further exploration.”

6. Speculate why in Figure 1G MP1 has worse survival than MP2.

Reply: We thank the reviewer for the comment. In this study, we analyzed the survival difference of non-metastatic patients in different MPs (shown in Figure 1G), aiming to support their different potential metastatic risk. Indeed, only MP-III had significantly shorter OS than MP-II and MP-I (MP-III vs. MP-II: log-rank $P < 0.0001$; MP-III vs. MP-I: log-rank $P < 0.0001$), while there was no significantly different OS between MP-I and MP-II (log-rank $P = 0.4947$). Similar results were observed that there was no significant difference in OS between MP-I and MP-II in five independent datasets, but there was shorter OS tendency of MP-II than that of MP-I in three datasets. To demonstrate the results more clearly, we have performed survival analysis for each of the two phenotypes. The results have been demonstrated in the Figures (Fig. 1G and Fig. 3) and described more clearly in the revised Results section on page 7: “Furthermore, we extracted the stage I treatment-naive patients and found that MP-III had significantly shorter overall survival (OS) than MP-I and MP-II (MP-III vs. MP-I: log-rank $P < 0.0001$; MP-III vs. MP-II: log-rank $P < 0.0001$, Fig. 1G), while there was no significantly different OS between MP-I and MP-II (log-rank $P = 0.4947$, Fig. 1G).” and

page 9: “Survival analysis (Fig. 3F-I) showed the stage I patients with MP-III to have significantly shorter OS than those with MP-I and/or MP-II in the GSE31210 (MP-III vs. MP-I, log-rank $P < 0.0001$, HR = 3.00, 95% CI = 1.56–5.76; MP-III vs. MP-II: log-rank $P = 0.0015$, HR = 6.34, 95% CI = 1.72–23.44), GSE50081 (MP-III vs. MP-I, log-rank $P = 0.0174$, HR = 1.86, 95% CI = 1.08–3.20; MP-III vs. MP-II: log-rank $P = 0.0043$, HR = 3.85, 95% CI = 1.42–10.39), GSE13213 (MP-III vs. MP-I, log-rank $P = 0.0412$, HR = 1.66, 95% CI = 1.00–2.76; MP-III vs. MP-II: log-rank $P = 0.4347$, HR = 1.58, 95% CI = 0.50–4.97), and GSE42127 datasets (MP-III vs. MP-I, log-rank $P = 0.0243$, HR = 2.31, 95% CI = 1.02–5.26; MP-III vs. MP-II: log-rank $P = 0.0754$, HR = 2.64, 95% CI = 0.86–8.10). In the GSE68465 dataset, MP-III also exhibited shorter OS but without statistical significance (log-rank $P > 0.05$, Fig. 3J), which might be caused by the lack of one gene (*PGM2*) for clustering in the dataset. Additionally, there was no significant difference in OS between MP-I and MP-II in five independent datasets, but there was shorter OS tendency of MP-II than that of MP-I in three datasets (GSE13213, GSE42127, and GSE68465), indicating that MP-II was an unstable metabolic pattern with fluctuations in OS.”

Additionally, the correlation analysis of three MPs with metastatic status in TCGA showed that MP-II was significantly enriched with lymph node metastatic samples. Currently, the lymph node dissection after surgery is the common treatment strategy for LUAD, and could improve survival of patients with high risk of lymph node metastasis, which might be the reason for the similar survival between patients in MP-II and MP-I. In accordance with your suggestion, we also mentioned this issue in the revised Discussion section on page 17: “The MPs could be confirmed in multiple independent cohorts, and the stage I LUAD patients with MP-III exhibited the worst prognosis and the highest risk of developing distant metastasis after surgery. In contrast, MP-II was inclined to be enriched with lymph node metastatic samples, while it was not observed to have a significantly poorer prognosis than MP-I. Currently, the lymph node dissection after surgery is the common treatment strategy for LUAD to improve survival of patients with high risk of lymph node metastasis [29], which might be the reason for the similar survival between patients in MP-II and MP-I.”

7. It is unclear why (lines 156-159) MP1, 2 and 3 contain similar met samples and recurrent samples. First of all, they should explain what do they mean by “recurrent”. Subsequently they should explain how the above makes sense with MP3 being more “severe” than MP1.

Reply: We apologize for the unclear presentation of samples with different metastases among the three MPs. The percentage of samples with different metastases status in each phenotype was added in revised Figure S2B, which was more suitable for reflecting the difference. Although the numbers of metastatic samples were similar in the three phenotypes, the percentage of samples with distant metastasis in MP-III was

significantly higher than that in MP-I ($P = 0.0169$) because the sample number in MP-I ($n=29$) is nearly three times than that of MP-III ($n=10$). To present the result more clearly, we have revised the description on page 7: “Similarly, the percentages of distant metastatic samples were significantly different among the three phenotypes in GSE11969 (Fisher’s exact test, $P = 0.0169$, Fig. S2A-B), which was the highest (50%) in MP-III and lower in MP-II (19.61%) and in MP-I (10.34%).”

We apologize for the unclear description about the recurrent information of samples. In the GSE11969 dataset, ‘recurrent’ represents the patients with in situ relapse after surgery. In correspondence with the sample information in the discovery dataset (TCGA), we re-extracted the patients with distant metastasis, lymph node metastasis, or non-metastasis in the GSE11969 and re-validated the association of MPs with metastasis. We have described the selection of samples in the dataset more clearly in the revised Methods section on page 21: “From the public databases, we collected two metastasis datasets (TCGA and GSE11969) that met the inclusion criteria of including LUAD patients with distant metastases, lymph node metastasis and non-distant metastases.” and “For the GSE11969 dataset, an independent validation dataset for metastasis, we extracted the mRNA expression profiles of 18 distant metastatic, 15 lymph node metastatic, and 35 non-metastatic samples.”

In the TCGA and GSE11969 datasets, we found that the distant metastatic samples were enriched in MP-III whereas non-metastatic samples were enriched in MP-I and lymph node metastatic samples were inclined to enriched in MP-II, suggesting a stronger tendency of distant metastasis of MP-III compared with the other two MPs. Furthermore, even in patients with stage I, the patients in MP-III exhibited significantly shorter OS than the patients in MP-II and MP-I, supporting the MP-III being more “severe” than other two phenotypes. In response to this comment, we have described the results more clearly and proposed the inference in the revised Results section on page 7: “We found MP-I, MP-II, and MP-III to be significantly enriched in non-metastatic, lymph node metastatic, and distant metastatic samples, respectively (Fisher’s exact test, $P < 0.0001$, Fig. 1F). Furthermore, we extracted the stage I treatment-naive patients and found that MP-III had significantly shorter overall survival (OS) than MP-I and MP-II (MP-III vs. MP-I: log-rank $P < 0.0001$; MP-III vs. MP-II: log-rank $P < 0.0001$, Fig. 1G), while there was no significantly different OS between MP-I and MP-II (log-rank $P = 0.4947$, Fig. 1G). Similarly, the percentages of distant metastatic samples were significantly different among the three phenotypes in GSE11969 (Fisher’s exact test, $P = 0.0169$, Fig. S2A-B), which was the highest (50%) in MP-III and lower in MP-II (19.61%) and in MP-I (10.34%). Whereas the percentages of lymph node metastatic samples were not observed to be significantly different among the three phenotypes (Fisher’s exact test, $P > 0.05$). The results indicated that patients in MP-III might have the strongest potency for distant metastasis with poor prognosis.”

8. Explain why they selected GSE11969 for validation (line 154)? Why only such dataset? If other dataset are utilized, how reproducible are their data?

Reply: We are sorry for the confusion of selection of the GSE11969. Previously, we collected the gene expression datasets that recorded the patients occurring distant metastasis, lymph node metastasis or not when they were at diagnosed. We have continued to search the gene expression profiles in the public databases (GEO, ArrayExpress etc), while there was no other dataset met the inclusion criteria for metastasis validation. We have described the inclusion criteria for the gene expression datasets used in this study more clearly in the revised Methods on page 21: “From the public databases, we collected two metastasis datasets (TCGA and GSE11969) that met the inclusion criteria of including LUAD patients with distant metastases, lymph node metastasis and non-distant metastases.”

Additionally, we found that the GSE13213 dataset, used for survival analysis, also recorded the follow-up metastasis information after surgery, which is valuable for validating our inference. According to the reviewer's suggestion, we have utilized the GSE13213 dataset for further metastasis validation, and showed that 22.5% stage I patients in MP-III developed distant metastasis after surgery, marginally significantly higher than that (6.67%) in MP-I (Fisher's exact test, $P = 0.0513$), and a little higher than that (12.5%) in MP-II but without significance. We have supplemented the result in the revised Results section on page 10: “Especially, the GSE13213 dataset also recorded the follow-up metastasis information until five years after surgery (Fig. S2C-D), which showed that 22.22% stage I patients in MP-III developed distant metastasis after surgery, marginally significantly higher than that (6.67%) in MP-I (Fisher's exact test, $P = 0.0513$) and tentatively higher than that (12.50%) in MP-II (Fisher's exact test, $P = 0.66$).” and described the follow-up metastasis information in revised Methods section on page 21: “Therein, GSE13213 dataset also recorded the follow-up metastasis information of the stage I patients after surgery within five years, among which nine patients developed distant metastases, one patient had lymph node metastasis and 52 patients did not have any metastasis.”

9. In describing Figure 2 they mention MP1 and MP3 data, what about MP2? Everything that is presented in Figures must be described and discussed.

Reply: In the previous version of the manuscript, we analyzed the differences across MP-I, -II and -III, while we mainly focused on the descriptions of molecular characteristics of MP-III and MP-I, which had the highest and lowest risk of metastasis. According to the reviewer's suggestion, we have described all the results presented in Figure 2 more clearly in the revised Results section on page 7: “The percentage of each stage in each MP was calculated, and showed that different stages had significantly different percentages among the three phenotypes (Fisher's exact test, $P < 0.0001$); the

percentages of stage III (30.99%) and stage IV (15.49%) in MP-III was obviously higher than that in the other two MPs, the percentage of stage II (33.83%) was the highest in MP-II, whereas the percentage of stage I (58.15%) was the highest in MP-I. Analogously, we calculated the percentage of each transcriptional subtype [14] in each MP, and found that the percentages of magnoid subtype, squamoid subtype, and bronchioid subtype were the highest in MP-III (80.28%), MP-II (27.94%) and MP-I (66.31%), respectively (Fisher's exact test, $P < 0.0001$). This is consistent with current knowledge that MP-I was enriched with bronchioid subtype, which is characterized by a low grade and the least tumor cell invasion [15]. We further calculated molecular scores based on mRNA expression profiles (see Supplementary Information) and found the hypoxia score [16] to increase progressively from MP-I to MP-III through MP-II (Wilcoxon rank test, $P < 0.0001$). Similar results were obtained for the stemness [17] and proliferation scores [18] (Wilcoxon rank test, $P < 0.0001$). The immune score was the highest in MP-II, whereas the lowest was in MP-III (Wilcoxon rank test, $P < 0.0001$). In contrast, the tumor mutation burden (TMB) was observed to be highest in MP-III, while decreased to MP-II and MP-I (Wilcoxon rank test, $P < 0.0001$)."

10. MP3 is associated to TP53, KEAP and SETD2 anomalies. Why do they talk about "4 genes"? They say "MP3 have lesions in at least one of the 4 genes". TMB is not a gene. In addition, what is the relevance that 87.3% of MP3 have at least one mutation in those genes?

Reply: We are sorry for the mistake. Here, we found three genomic lesions (*TP53*, *KEAP1* and *SETD2*) that were associated with MP-III. We have revised the mistake in the revised Results.

In the previous version of the manuscript, we aimed to present the total mutation frequency of these genes covering MP-III to highlight the importance of these genes for the phenotype. Considering the reviewer's comment, we realized that it was not enough to support their importance, thus we removed this sentence for reducing the confusion. Instead, we analyzed the correlations of the three genetic events with the mRNA expression of the 11 metabolic genes to elucidate their roles in regulating metabolic reprogramming, which has been described in the revised Results section on page 9: "The correlation analyses of the genetic events with the mRNA expression of the 11 metabolic genes showed that *TP53* mutation could regulate mRNA expression of all other metabolic gene except a glycolytic gene (*LDHA*) and a lipid metabolic gene (*PLPPI*). Additionally, mutated *KEAP1* could regulate four glycolytic genes (*TP11*, *ALDOA*, *GAPDH*, *GPI*) and two lipid metabolic gene (*PLD3*, *GPD1L*), meanwhile, the mRNA expression of two glycolytic genes (*TP11*, *PGM2*) and another lipid metabolic gene (*GPD1L*) were significantly associated with deleted *SETD2* (Pearson correlation, FDR < 0.05)."

11. Table S2 require % not only numbers. Stage IV has equal distribution in MP1 and MP2 and it is just slightly higher in MP3; this is not convincing. Only stage 2 is higher in MP2 (versus what? Significance versus MP1 or MP3? It really is not clear). They mention in text that MP3 is enriched in squamoid (not aquamoid, as written) and magnoid. Actually, MP2 is enriched in those subgroups, according to the figure. It is all presented in a very confusing and not carefully organized manner. Also, bronchioid is not enriched in MP1 as stated in text, but it is in MP3, if I am interpreting correctly suppl table2. They should explain differences in TMB and immune score, it is unclear why the first parameter is high and the second low in MP3.

Reply: In according with your suggestion, we demonstrated the percentages of subgroups for each clinical and molecular characteristic across each MP in the revised Supplementary Table 2. In this study, we compared the differences in percentages of each factor among the three MP using Fisher's exact test under the null hypothesis that the percentages at the row variables (clinical and molecular characteristics) are the same for different values of other column variables (MPs). We have revised the Methods section on page 24: "Fisher's exact test was performed to evaluate whether the percentages of stages, transcriptomics subtypes [14] and frequencies of genetic lesions were significantly different across the multiple groups, under the null hypothesis that the percentages at the row variables are the same for different values of other column variables." The statistical test showed that there was significantly different percentages of different stages and transcriptional subtypes among the three MPs. Although the sample amount of stage IV in MP-III was only slightly higher than that in MP-I and MP-II, the percentage of stage IV (15.49%) in MP-III was significantly higher than that in MP-I (3.80%) and MP-II (5.26%) as the sample number of MP-III (n=71) was markedly less than that of MP-I (n=187) and MP-II (n=136). Analogously, there was significantly different percentages of transcriptional subtypes between different MPs (Fisher's exact test, $P < 0.0001$) that the percentage of magnoid subtype was the highest in MP-III, the percentage of squamoid subtype (we have corrected the spelling) was the highest in MP-II and the percentage of bronchioid subtype was highest in MP-I. We have revised the description more clearly in the revised Results section on page 7: "The percentage of each stage in each MP was calculated, and showed that different stages had significantly different percentages among the three phenotypes (Fisher's exact test, $P < 0.0001$); the percentages of stage III (30.99%) and stage IV (15.49%) in MP-III was obviously higher than that in the other two MPs, the percentage of stage II (33.83%) was the highest in MP-II, whereas the percentage of stage I (58.15%) was the highest in MP-I. Analogously, we calculated the percentage of each transcriptional subtype [14] in each MP, and found that the percentages of magnoid subtype, squamoid subtype, and bronchioid subtype were the highest in MP-III (80.28%), MP-II (27.94%) and MP-I (66.31%), respectively (Fisher's exact test, $P < 0.0001$). This is consistent with current knowledge that MP-I was enriched with bronchioid subtype, which is characterized by a low grade and the least tumor cell invasion [15]. We further calculated molecular scores

based on mRNA expression profiles (see Supplementary Information) and found the hypoxia score [16] to increase progressively from MP-I to MP-III through MP-II (Wilcoxon rank test, $P < 0.0001$). Similar results were obtained for the stemness [17] and proliferation scores [18] (Wilcoxon rank test, $P < 0.0001$). The immune score was the highest in MP-II, whereas the lowest was in MP-III (Wilcoxon rank test, $P < 0.0001$).”

In accordance with the reviewer’s suggestion, we have explained the possible reason of the differences in TMB and immune score in the revised Discussion section on page 16: “Notably, MP-III was characterized with the highest TMB but the lowest immune score, indicating an immunologically ‘cold’ phenotype, which has been reported in NSCLC [26] and melanoma [27]. Previous studies have found that metabolic dysregulation of tumor cells could suppress the infiltration of immune cells and functions of antitumor immunity [28], resulting in the high TMB tumor translating into an immunologically ‘cold’ tumor [27]. Our further genetic analysis revealed that the mutations of *TP53* and *KEAP1*, and deletion of *SETD2* might be underlying the metabolic reprogramming. The exploration of the genetic responsibility for MP-III would guide us in designing the studies of clinical drug combination that may succeed in transforming these patients into those responsive to immunotherapy [26].”

12. Explain why the 2 other dataset (I and J) in Figure 3 do not show differences in OS. Which differences are there in these 2 datasets that make up for that observation? It is rather odd that MP3 stands out in F and G, that according to K and L, only contain stage I and II tumors. My understanding is that MP3 should contain more aggressive stages...

Reply: We thank the reviewer for the insightful comment. Our results indeed showed that MP-III contain more aggressive stages, while there was still a certain percentage of early stage (even stage I) patients without any metastasis clustered into MP-III. Thus, we inferred that this kind patient might have high-risk of metastasis during the development, resulting in poor prognoses. In order to validate the inference, we performed the survival analysis for patients without any metastasis (N_0M_0) among the three MPs. We have explained the reason in the revised Results section on page 9: “The above result showed that a certain percentage of early-stage patients without any metastasis could be clustered into MP-III; thus, we inferred that these patients might have high-risk of metastasis during the development, resulting in poor prognoses.”

According to your suggestion proposed in comment 13, we extracted stage I patients for survival analyses to exclude the influence of different stages among the three phenotypes to support the inference. The new results showed that except the three datasets, the GSE42127 dataset also supported the significant difference in OS (MP-III vs. MP-I, log-rank $P = 0.0243$, HR = 2.31, 95% CI = 1.02–5.26; MP-III vs. MP-II: log-rank $P = 0.0754$, HR = 2.64, 95% CI = 0.86–8.10, Fig. 3I). In the GSE68465 dataset, MP-III also exhibited the similar OS with MP-II, and shorter OS than MP-I but without

statistical significance (log-rank $P > 0.05$), which might be caused by the lack of one gene (*PGM2*) for clustering in the dataset. We have mentioned this issue in the revised Results section on page 9: “Survival analysis (Fig. 3F-I) showed the stage I patients with MP-III to have significantly shorter OS than those with MP-I and/or MP-II in the GSE31210 (MP-III vs. MP-I, log-rank $P < 0.0001$, HR = 3.00, 95% CI = 1.56–5.76; MP-III vs. MP-II: log-rank $P = 0.0015$, HR = 6.34, 95% CI = 1.72–23.44), GSE50081 (MP-III vs. MP-I, log-rank $P = 0.0174$, HR = 1.86, 95% CI = 1.08–3.20; MP-III vs. MP-II: log-rank $P = 0.0043$, HR = 3.85, 95% CI = 1.42–10.39), GSE13213 (MP-III vs. MP-I, log-rank $P = 0.0412$, HR = 1.66, 95% CI = 1.00–2.76; MP-III vs. MP-II: log-rank $P = 0.4347$, HR = 1.58, 95% CI = 0.50–4.97), and GSE42127 datasets (MP-III vs. MP-I, log-rank $P = 0.0243$, HR = 2.31, 95% CI = 1.02–5.26; MP-III vs. MP-II: log-rank $P = 0.0754$, HR = 2.64, 95% CI = 0.86–8.10). In the GSE68465 dataset, MP-III also exhibited shorter OS but without statistical significance (log-rank $P > 0.05$, Fig. 3J), which might be caused by the lack of one gene (*PGM2*) for clustering in the dataset. Additionally, there was no significant difference in OS between MP-I and MP-II in five independent datasets, but there was shorter OS tendency of MP-II than that of MP-I in three datasets (GSE13213, GSE42127, and GSE68465), indicating that MP-II was an unstable metabolic pattern with fluctuations in OS. Especially, the GSE13213 dataset also recorded the follow-up metastasis information until five years after surgery (Fig. S2C-D), which showed that 22.22% stage I patients in MP-III developed distant metastasis after surgery, marginally significantly higher than that (6.67%) in MP-I (Fisher’s exact test, $P = 0.0513$) and tentatively higher than that (12.50%) in MP-II (Fisher’s exact test, $P = 0.66$).”

13. Also, what is the value/relevance of classifying tumors in MP subpopulations with different levels of aggressiveness, if we need staging to get differences in OS?

Reply: We thank the reviewer for the insightful suggestion. The relevance analysis showed that MPs were significantly associated with clinical stages, as mentioned in our response to comment 11. According to reviewer’s suggestion, we aimed to perform the survival analysis in each stage. However, limited by the small sample numbers of stage II and III patients in all datasets, we only studied stage I patients for the survival analysis to exclude the influence of aggressive stages, as mentioned in our response to comment 12.

14. Figure 4D: not clear how the level of significance can be so high in the lipid panel

Reply: We thank the reviewer for the comment. The Wilcoxon rank test showed significant differences in expression of the lipid metabolic genes among the three MPs, and the high level of significance might be caused by the large numbers of samples (cells) used for comparison. While, the boxplot (Fig. 4D) did not show clearly, which might be

caused by the high percentages of zero value of the lipid metabolic genes in the single-cell sequencing dataset.

15. I assume by gene expression they mean RNA expression (line 216)?

Reply: We are sorry for this confused description. Here, we investigated a correspondence between the mRNA expression of each gene with its protein expression using Pearson correlation, and found that the mRNA expression of all the 11 genes was significantly and positively correlated with the expression of the corresponding proteins. We have described more clearly in the revised Results section on page 10: “**Moreover, we found that the mRNA expression of each metabolic gene was significantly positively correlated with its protein expression in the dataset (Pearson correlation, FDR < 0.05, Fig. 4B).**”

16. What does the PET radiomic dataset add?

Reply: We apologize for the unclear description of the purpose of the PET radiomic dataset. Currently, the levels of some metabolic parameters, such as the maximum standardized uptake value (SUVmax), metabolic tumor volume (MTV), and total lesion glycolysis (TLG), were used to validate the metabolic abnormality of tumor cells reflected by PET-CT imaging, especially glucose metabolism. Thus, in this study, we aimed to utilize PET-CT radiogenomic data to support the difference in metabolic status among the three MPs. Based on the reviewer's comment, we reconsidered that the result of PET radiomic dataset tended to represent glucose metabolism rather than discoordination of glycolysis-lipid metabolism. Therefore, we removed this result in the revised manuscript.

17. Explain SUVmax, MTV, and TLG levels when mentioning them

Reply: Done as suggested, which was mentioned in our response to comment 16.

18. Clusters are not described well: cluster 1 is a mix of normal and tumor; Normal is also contained in a small subcluster of cluster 5

Reply: We apologize for the unclear description about the trajectory states. Although State 1 was a mixture of normal epithelial cells and primary tumors epithelial cells, the trajectory analysis showed that more percentage of normal epithelial cells located on the initiate of State 1, gradually decreasing to the end of the state. In contrast, the less primary tumors epithelial cells located on the initiate of State 1, gradually increasing to

the end of the state. The "pseudo-time" result, presented in Figure 4F, indicated that the normal cells gradually "differentiated" into primary tumor epithelial cells along with the changes in the mRNA expression of the 11 metabolic genes. We have described the result more clearly in the revised Results section on page 12: “Here, the metabolic trajectory of the glycolysis-lipid imbalance appeared to begin principally from partial normal epithelial cells, which marked the beginning of State 1, evolved to MP-I tumor cells majorly at the end of State 1, and then formed a branched structure with two major cell fates (Cell fate 1 and 2) based on the changing in mRNA expression of the 11 metabolic genes (Fig. 5F).”

Here, there were a small percentage (0.89%) of normal epithelial cells in State 5. The scRNA-seq has emerged as a valuable tool to study the intrinsic heterogeneity in cellular composition within tumors and normal tissues, which might not be discovered by bulk dataset. Therefore, we considered that some cells in the tumor-adjacent tissue might exhibit some characteristics like tumor cells, such as metabolic dysregulation [2, 3], which were discovered by scRNA-seq dataset. This might be one of the reasons that can be used to explain the small percentage of normal epithelial cells in State 5, which is an intermediate metabolic state linking State 1 to State 6 and State 7. Additionally, the phenomenon might be caused by the mistake in assessing by the scRNA-seq algorithms, thus we did not discuss this phenomenon in the manuscript.

19. The lipidic genes are the same in normal and brain mets... really a signature of 3 genes is not so realistic to adopt

Reply: The Wilcoxon rank test showed that the average mRNA expression of three lipid metabolic genes were significantly higher in the normal cells than that in the brain metastatic cells ($P < 0.0001$), while the median of boxplot (Fig. 5B) did not show clearly, which might be caused by the high percentages of zero value of the lipid metabolic genes in the scRNA-seq dataset. As the reviewer said, three genes might be too few to be adopted in clinic. In this study, we utilized the imbalance of the mRNA expression between the eight glycolytic genes and three lipid metabolic genes to characterize the metabolic reprogramming phenotypes associated with metastasis, which might be more robustness than 3 lipid metabolic genes. Additionally, according to your suggestion proposed in comment 5, we analyzed other glycolytic and lipid metabolic genes in corresponding biological pathways among three MPs and found more glycolytic and lipid metabolic genes to be significantly correlated with MPs, suggesting that the dysregulated patterns of the 11 metabolic genes might reflect the dysregulations in the two kinds of pathways. However, whether the definition of MPs might be improved by more glycolytic and lipid metabolic genes still need to be explored. We have mentioned this issue in the revised Discussions on page 16: “Additionally, we also found that a lot of other glycolytic and lipid metabolic genes in corresponding biological pathways were

significantly positively and negatively correlated with MPs (Pearson correlation, FDR < 0.05, Table S6), such as a key glycolytic enzyme (*PKM*) gradually increasing from MP-I to MP-III. The results indicated the stronger dysregulations of the two kinds of pathways in MP-III. However, whether more glycolytic or lipid metabolic genes could be used to improve the definition of MPs needs further exploration.”

20. Figure 4c: here the branching describing the clustering is different than in previous figures. Here MP2 and MP3 are coming out from the same branch and MP1 is separate. Instead in the previous figures, MP1 clustered closer to MP2, and MP3 was separate. How to interpret that?

Reply: We thank the reviewer for the comment. The branch of clustering performed differently in different datasets. For example, in training TCGA and further scRNA-seq datasets, MP-II and MP-III are coming out from the same branch and MP-I is separate, whereas MP-II clustered closer to MP-I, and MP-III is separate in other five datasets. Considering the fluctuations of expression pattern and survival of MP-II, we inferred that MP-II might be an intellectual phenotype between MP-I and MP-III, which has been mentioned in the Discussion section on page 10: “Additionally, there was no significant difference in OS between MP-I and MP-II in five independent datasets, but there was shorter OS tendency of MP-II than that of MP-I in three datasets (GSE13213, GSE42127, and GSE68465), indicating that MP-II was an unstable metabolic pattern with fluctuations in OS.”

21. Rather than association with MP3 they observed association with the glycolysis genes.

Reply: We thank the reviewer for the insightful comment. To get rid of the confusion, we have additionally analyzed the association of the glycolytic genes with the clinical outcome, which has been described in the revised Discussion section on page 15: “Notably, in the TCGA dataset, we tried to utilize the eight glycolytic genes to cluster the samples into two phenotypes, but there was no significant difference in stage I patients’ OS between the high- and low-glycolysis groups (log-rank $P = 0.66$, Fig. S5A-B). Next, we extracted the patients with high expression pattern of glycolytic genes for further hierarchical clustering based on the three lipid metabolic genes, and found that the patients with low expression pattern of lipid metabolism had significantly shorter OS than the others (log-rank $P < 0.0001$, Fig. S5C-D). These results suggested that high glycolytic and low lipid metabolic levels might synergistically accelerate tumor progression in LUAD.”

22. Interpretation of Trajectory is not convincing. How can you find normal cells at the tip of a trajectory adopted to explain how tumors evolve? If they suspect there was noise during

single cell sequencing they should disregard all the data. How do they exclude their other data are not “contaminated” by noise? Description of Fig 4G is extremely poor.

Reply: We are sorry for the unclear description of the results concluded from the Trajectory. In the TCGA bulk expression dataset, compared with the normal tissues, the tumor tissues showed significantly higher expression of glycolytic genes, lower expression of lipid metabolic genes, and gradually enhanced dysregulation from MP-I to MP-III. The result suggested that the normal cells might be the tip of the metabolic ‘evolution’ of tumors. Therefore, we performed the trajectory analysis for the epithelial cells derived from normal and tumor tissues based on the 11 metabolic genes. As expected, the result showed that MP-I tumor cells were the closest to the normal cells, and then differentiated into MP-II and MP-III. In order to better understand the trajectory analysis, we have described the inference in the revised Results section on page 11: **“In the TCGA dataset, the tumor cells exhibited significantly higher expression of glycolytic genes and lower expression of lipid metabolic genes than normal cells, and the dysregulation gradually increased from MP-I to MP-III. Therefore, we inferred that the normal cells might be the tip of the metabolic evolution of tumors, and differentiated to MP-I and then to MP-II and MP-III.”**

The trajectory analysis showed that these epithelial cells were classified into seven different states, among which State 1 and State 4 were enriched in normal epithelial cells. Theoretically, tumor epithelial cells arise from normal cells; thus State 1 and State 4 should be both treated as the origins for the evolution of these cells. However, we found that the normal cells in State 4 were characterized by very low expression of all the 11 metabolic genes, which were not observed in the bulk gene expression profiles. In contrast, the normal cells in State 1 exhibited the low expression of glycolytic genes and high expression of lipid metabolic genes, consistent with the phenomenon observed in the bulk gene expression profiles. Therefore, we previously inferred that State 4 might be “contaminated” by noise, which is a quite arbitrary and not convincing. In response to this comment, we compared the cells in State 4 and the cells in other states using the single-cell differential expression analysis method (DEsingle package), and found that the expression pattern of the 11 DE genes in State 4 was still stable after removing the influence of noise, indicating that State 4 was not “contaminated” by noise. Thus, we considered that the cells in this state may be a “low-low” phenotype of glycolysis and lipid metabolism, as discovered in melanoma cells [4], while it was not discovered in the bulk expression profiles. As the phenotype was beyond the focus of this study, we removed the cells from the further analyses, and discussed this issue in the revised Results section on page 12: **“Notably, State 1 and State 4 were both enriched with normal epithelial cells, which should be treated as the origins of the evolution. We found that the cells in State 1 exhibited the low mRNA expression of glycolytic genes and high mRNA expression of lipid metabolic genes, whereas State 4 exhibited very low expression of all the 11 metabolic genes, which was not observed in the bulk dataset**

(TCGA). Excluding the cause of noise for State 4 by DEsingle method, we considered that the cells in the state might be a unique “low-low” metabolism phenotype discovered by scRNA-seq dataset, which was not analyzed in the further analyses as it was beyond this study. Here, the metabolic trajectory of the glycolysis-lipid imbalance appeared to begin principally from partial normal epithelial cells, which marked the beginning of State 1, evolved to MP-I tumor cells majorly at the end of State 1, and then formed a branched structure with two major cell fates (Cell fate 1 and 2) based on the changing in mRNA expression of the 11 metabolic genes (Fig. 5F). Tracing the metabolic trajectory of Cell fate 1 (ignoring State 4) revealed that MP-I tumor cells (State 2) evolved into MP-III tumor cells (State 3).” and revised Discussion section on page 17: “Notably, the evolutionary trajectory exhibited a new “low-low” metabolic phenotype of normal and tumor epithelial cells in the scRNA-seq dataset, which has been reported in melanoma [27]. The unique phenotype was not discovered in the bulk expression profiles of LUAD, meriting further validation and exploration.”

23. Fig 4I: same colors for state 7 and glycolysis

Reply: We are sorry for this mistake that we have revised in the figure.

24. Interactome: the immune cells can be subdivided further, plus what is the lymphocyte activation state? The TME contribution may contain less active immune cells, also depending from tumor stage....

Reply: According to the reviewer’s helpful suggestion, we have reanalyzed the intercellular communications, and decoded more sophisticated interactome of MPs with refined cell subtypes defined in the original article. Since the results were similar with previous results, we have discussed this result in the revised Discussion section on page 18: “We also decoded more sophisticated interactome of the MPs with refined subdivided cell subtypes (Fig. S7, Table S7), such as activated DCs and Cytotoxic CD8⁺ T lymphocytes etc., and observed the similar results. For example, MP-III still exhibited a unique interaction with the refined endothelial cells (Lymphatic ECs, Stalk-like ECs, Tip-like ECs and Tumor ECs) on ANGPTL signaling pathway, indicating the important role of MP-III population in promoting angiogenesis of all the endothelial cells. We additionally observed that Cytotoxic CD8⁺ T and Exhausted CD8⁺ T lymphocytes to be sources of *IFNG* in IFN-II signaling pathway that acted on MP-II and MP-III populations (Fig. S7). Notably, the interaction intensity of the Cytotoxic CD8⁺ T lymphocytes with MP-II was stronger than that with MP-III, consisting with the phenomenon observed in the bulk dataset (TCGA) that MP-II had the highest immune score.” In this study, we performed the interactome analysis in the early stage LUAD patients to reveal the specific intercellular communications of MPs with other cells,

which might promote metastasis of some primary tumor cells.

25. Interactome Fig 5b: describe and speculate better on the data. They say almost nothing about all the data, they just cannot say that MP2 and MP3 talk more...what are the different contribution of all the subpopulation they show? Biological relevance or at least hypothesis on the observations?? The entire interactome is just speculative. As presented it is not a useful information.

Reply: We thank the reviewer for the comment. The hypothesis underlying the interactome was that the metabolic reprogramming allows cancer cells to migrate away from primary tissues by altering their microenvironment. Previously, we aimed to provide the evidence of the stronger entire interactome of MP-II and MP-III with other microenvironment cell populations, which was presented in Fig.5B. However, as the reviewer said, the different contribution to the subpopulation they show might provide more useful information, which was shown in Fig. 5A and 5D. Therefore, we have removed the result in Fig. 5B and added the hypothesis underlying the interactome as suggested in the revised Results section on page 13: “In accordance with the hypothesis that the metabolic reprogramming allows tumor cells to migrate away from primary tissues by altering the interactions with other cells in the tumor microenvironment, we build a metabolic phenotype-specific cell-cell communication atlas based on the 6,372 tumor epithelial cells (MP-I, -II, and -III), 2,373 stromal cells (fibroblasts and endothelial cells) and 35,506 immune cells (B lymphocytes, T lymphocytes, NK, myeloid, and MAST cells) extracted from the primary tissues of patients with tLung. In total, we identified 41 significant ligand-receptor pairs between the 10 cell populations (Fig. 6A), and as expected the MP-III population showed the strongest outgoing signal, followed by MP-II, with MP-I weakest (Fig. 6B).”

As the reviewer said, the cell-cell communications were speculative by ligand-receptor expression, thus the result needs special attentions. Our result showed that MP-III had the stronger intensity of interactions with endothelial cells in the VEGF signaling pathway, which supported our inference that MP-III had the highest metastasis potency, and was consistent with the reported biological phenomenon that the tumor cells with metabolic reprogramming could accelerate metastases by inducing angiogenesis of endothelial cells [5]. The result suggested that CellChat could identify some possible cell-cell communications. Additionally, in response to the comment, we validated these interactions in an independent dataset in the revised Results section on page 14: “Collectively, we identified 103 ligand-receptor pairs interacting MP-III with other cell populations (Table S4). Similarly, we identified 154 significant ligand-receptor pairs interacting MP-III with other cell populations in an independent scRNA-seq dataset (GSE123902). Thereinto, there was 62 overlapped ligand-receptor pairs, significantly higher than expected (Hypergeometric distribution model, $P < 0.0001$, Fig.

S4). Notably, mRNA expression of nine ligands expressed on MP-III, including *ANGPTL4*, *VEGFA* etc., were significantly higher than that on MP-I, however, none of them is known driver gene for LUAD. All nine ligands were validated in another scRNA-seq dataset (Wilcoxon rank test, FDR < 0.05, Table S5).”

Even so, we considered that the cell-cell communications inferred by ligand-receptor expression is indeed a limitation of current single-cell interactions, as the reviewer said, which was mentioned in the revised Discussion section on page 20: “Second, the cell-cell interactions discovered in this study should be carefully determined, as they were speculative by the mRNA expression of ligands and their receptors. Though the significantly more interactions of MP-III population with other populations were validated in another scRNA-seq dataset, the relative reliability of the communications still needs further experimental verification.”

26. Stratification of interactions with known mutation in epithelial cells?

Reply: In according with your suggestion, we found that none of the nine genes expressed on MP-III interacted with other cells was known ‘driver’ mutations for LUAD, which have described in the revised Results section on page 14: “Notably, mRNA expression of nine ligands expressed on MP-III, including *ANGPTL4*, *VEGFA* etc., were significantly higher than that on MP-I, however, none of them is known driver gene for LUAD.”

References

1. Cano-Corres R, Sanchez-Alvarez J, Fuentes-Arderiu X. The Effect Size: Beyond Statistical Significance. *EJIFCC*. 2012;23(1):19-23.
2. Kinker GS, Greenwald AC, Tal R, et al. Pan-cancer single-cell RNA-seq identifies recurring programs of cellular heterogeneity. *Nat Genet*. 2020;52(11):1208-18.
3. Xiao Z, Dai Z, Locasale JW. Metabolic landscape of the tumor microenvironment at single cell resolution. *Nat Commun*. 2019;10(1):3763.
4. Spranger S, Bao R, Gajewski TF. Melanoma-intrinsic beta-catenin signalling prevents anti-tumour immunity. *Nature*. 2015;523(7559):231-5.
5. Maishi N, Hida K. Tumor endothelial cells accelerate tumor metastasis. *Cancer Sci*. 2017;108(10):1921-6.

Reviewer #2 (Remarks to the Author):

In general I think this is good paper which focuses needed attention on the specific role of metabolic phenotypes in differentially enabling metastasis. Most of the results are reasonably convincing and hence I only have some technical requests/comments

- Can the authors be more specific as the criteria used to pick the 11 genes used for phenotype classification, from the larger set of differentially expressed genes.

Reply: We thank the reviewer for the comment. In this study, we found that the upregulated DE genes were significantly enriched in biological pathway related to glycolysis, while the downregulated DE genes were potentially enriched in lipid metabolism, suggesting the imbalance of glycolysis and lipid metabolism during metastasis. Previous studies showed that the discoordination of glycolysis and lipid metabolism of tumors to be associated with clinical outcome [1]. Therefore, we focused on the upregulated DE genes in the glycolysis and downregulated DE genes in the two enriched lipid metabolism pathways ('Glycerophospholipid metabolism' and 'Ether lipid metabolism'), to explore the metabolic reprogramming phenotypes. According to the reviewer's suggestion, we have described this process more clearly and named the 11 genes in the revised Results section on page 5: "Previous studies showed the discoordination of glycolysis and lipid metabolism of tumors to be associated with clinical outcomes [13]. Therefore, in this study, we focused on the upregulated genes involved in glycolysis and downregulated genes in lipid metabolism to analyze the metabolic reprogramming associated with LUAD metastases. Totally, we extracted the eight upregulated DE genes (*ALDOA*, *ENO1*, *GAPDH*, *GPI*, *LDHA*, *PGAMI*, *PGM2*, *TPII*) from the 'Glycolysis/Gluconeogenesis' pathway and three downregulated DE genes (*PLPPI*, *GPDIL*, *PLD3*) from the two lipid metabolism pathways ('Glycerophospholipid metabolism' and 'Ether lipid metabolism'). Here, the detailed functions of the 11 DE genes are represented in Table S1. Additionally, we analyzed the mRNA expression correlation of these genes and found that the 11 DE genes were significantly positively correlated with each other within the same kind pathway, while significantly negatively correlated with the genes in other kind pathways (Pearson correlation, $FDR < 0.05$, Fig. S1). These results indicated that the imbalance of the glycolysis and lipid metabolism might participate in LUAD metastasis."

-Could the authors provide some more information about the method used to infer transcriptional trajectories (p. 14). The results of this section I found to be less convincing than other aspects of the paper. For example, is there any way to distinguish gene dropouts (line 258) from the possibility the some cells exhibit a low metabolic phenotype (as seen in other situation, for example Jia, Dongya, et al. "Drug-tolerant idling melanoma cells exhibit

theory-predicted metabolic low-low phenotype." *Frontiers in oncology* 10 (2020): 1426.). And, is there an overall picture that one can be certain of, given the murkiness of the findings in Fig 4

Reply: We are sorry for the unclear description about transcriptional trajectories. We utilized the reverse graph embedding machine learning algorithm method to describe multiple fate decisions in a fully unsupervised manner and obtain the evolution trajectory of epithelial cells. Previous result showed that metabolic reprogramming might be changed during tumor cell progression, thus, we determined distinct cell state based on the changes in mRNA expression of the 11 metabolic genes to learn a principal graph that represented the cell metabolic trajectory. We have described more clearly in the revised Method section on page 25: “Cell trajectory and branched expression analyses were performed based on Monocle (version 2), using the reverse graph embedding machine learning algorithm to learn the changes in 11 metabolic gene expression sequences that each cell must go through as part of a dynamic biological process (here, metabolic reprogramming). The dimensionality of the cells was reduced by the DDRTree method, sequenced in pseudotime trajectory, and finally visualized [52].”

We thank the reviewer for the insight comment. We have analyzed the differential expression of the 11 genes between State 4 and other states using DEsingle package, and found that 11 DE genes expression pattern in State 4 were still stable after removing the influence of noise, indicating State 4 was not caused by the noise. Thus, we considered that the cells in this state may be a "low-low" phenotype of glycolysis and lipid metabolism, as discovered in the melanoma [2], while it was not discovered in the bulk expression profiles. As the phenotype was beyond the focus of this study, we removed the cells from the further analyses, and described in the revised Result section on page 12: “Notably, State 1 and State 4 were both enriched with normal epithelial cells, which should be treated as the origins of the evolution. We found that the cells in State 1 exhibited the low mRNA expression of glycolytic genes and high mRNA expression of lipid metabolic genes, whereas State 4 exhibited very low expression of all the 11 metabolic genes, which was not observed in the bulk dataset (TCGA). Excluding the cause of noise for State 4 by DEsingle method, we considered that the cells in the state might be a unique “low-low” metabolism phenotype discovered by scRNA-seq dataset, which was not analyzed in the further analyses as it was beyond this study.”

-I think it would be useful to present some more discussion of the role of the specific genetic modifications as being partially responsible for the metabolic changes. For example, KEAP is known to be an inhibitor of NRF2 and hence can control the level of ROS in cells; TP53 can affect the EMT status of cells and its malfunction can help drive a more glycolytic state and more metastatic competence

Reply: We thank the reviewer for the helpful comment. Previously, we have connected genomic lesions with metabolic genes through correlation analysis. According to the reviewer's suggestion, we supplemented protein-protein interaction between genomic lesions with the 11 metabolic genes to further demonstrate the possible interactions. And, we have tried to explain the role of the specific genetic modifications for regulating the metabolic changes in the revised Discussion section on page 19: “Finally, in order to comprehensively elucidate the underlying molecular alterations for metastasis, we recapitulated a metabolic phenotype-specific cell-cell communication atlas of LUAD and linked it with cell-autonomous oncogenic events through metabolic genes, thereby providing novel therapeutic targets. The comprehensive network demonstrated that *TP53* mutation, *KEAPI* mutation, and *SETD2* deletion could regulate the mRNA expression of these metabolic genes, bringing about the metabolic reprogramming. Recent studies have shown that mutations in *TP53* [37-39] and *KEAPI* [40] contribute to various metabolic disorders and play pivotal roles in metabolic reprogramming and cancer progression. Furthermore, the protein-protein interaction network from the STRING database (<https://string-db.org/>) indicated that *TP53* could activate glycolysis pathway through up-regulating *PGM2* [37], and hereby enhance the intercellular communication of MP-III with endothelial cells on the VEGF signaling pathway (*VEGFA-KDR*). Meanwhile, *TP53* could regulate lipid metabolism by interacting with *HIPK4* to *PLD3*. Previous studies have shown that the mutation in *KEAPI* could activate *NRF2*, directing epithelial cells toward metabolic reprogramming of glucose metabolism [40, 41]. In this study, we found that the mutation of *KEAPI* might indirectly interact with a glycolytic gene (*LDHA*) through *COPS5*, which is related to the activation of glycolysis [42]. Meanwhile, *KEAPI* might also affect lipid metabolism through indirectly interaction with *TP53*. The histone H3K36 methyltransferase *SETD2* is frequently mutated or deleted in a variety of human tumors [43-45]. Nevertheless, the role of *SETD2* deletion in promoting metastasis and metabolic reprogramming remains largely undefined. In this study, we found that *SETD2* deletion might participate in glycolysis-lipid metabolism discoordination by indirectly interacting with *TP53* and *KEAPI*.”

-Do the authors have any ideas as to how to directly verify the results using CELLCHAT?

I am somewhat skeptical that ligand-receptor expression levels are a reliable indicator of actual information transfer between cells and I would have liked some level of validation. Also, there are clearly other modes cell-cell communication, for example through metabolites, which cannot be captured by expression levels, A more complete discussion would be helpful here.

Reply: We thank the reviewer for the helpful comment. In according with your suggestion, we additionally validated these cell-cell communications in an independent

scRNA-seq data in the revised Results section on page 14: “Collectively, we identified 103 ligand-receptor pairs interacting MP-III with other cell populations (Table S4). Similarly, we identified 154 significant ligand-receptor pairs interacting MP-III with other cell populations in an independent scRNA-seq dataset (GSE123902). Thereinto, there was 62 overlapped ligand-receptor pairs, significantly higher than expected (Hypergeometric distribution model, $P < 0.0001$, Fig. S4). Notably, mRNA expression of nine ligands expressed on MP-III, including *ANGPTL4*, *VEGFA* etc., were significantly higher than that on MP-I, however, none of them is known driver gene for LUAD. All nine ligands were validated in another scRNA-seq dataset (Wilcoxon rank test, FDR < 0.05, Table S5).”

Additionally, we also discussed this issue that the cell-cell communications inferred by ligand-receptor expression is indeed a limitation of current single-cell interactions in the revised Discussion section on page 20: “Second, the cell-cell interactions discovered in this study should be carefully determined, as they were speculative by the mRNA expression of ligands and their receptors. Though the significantly more interactions of MP-III population with other populations were validated in another scRNA-seq dataset, the relative reliability of the communications still needs further experimental verification.”

According to the reviewer’s insightful suggestion, we have discussed the other important modes of metabolic reprogramming in altering the microenvironment in the revised Discussion section on page 20: “Furthermore, besides the interactions of ligands and receptors, the other modes of metabolic reprogramming in altering the microenvironment, such as metabolites, merit further exploration.”

References

1. Karasinska JM, Topham JT, Kalloger SE, et al. Altered Gene Expression along the Glycolysis-Cholesterol Synthesis Axis Is Associated with Outcome in Pancreatic Cancer. *Clin Cancer Res.* 2020;26(1):135-46.
2. Spranger S, Bao R, Gajewski TF. Melanoma-intrinsic beta-catenin signalling prevents anti-tumour immunity. *Nature.* 2015;523(7559):231-5.

Reviewer #3 (Remarks to the Author):

In this manuscript, Xin Li et al. conducted transcriptome analysis from lung adenocarcinoma and identified 11 genes which were related in three metabolic reprogramming phenotypes and associated with metastasis. They successfully classified patients in TCGA and independent

public datasets with transcriptome or proteome information. They also performed analysis of one scRNA-seq dataset and identified intercellular interaction which was related in metabolic phenotypes MP-I, MP-II and MP-III. There are some points that need to be addressed to improve the manuscript as below;

Points;

1. The authors extracted 11 DE metabolic genes from a lot of DE genes. I think that this step is one of the most critical steps in this study. Please describe the details how to select eight glycolysis and three lipid metabolism genes. Are there only 11 genes associated with these metabolic pathways in all DE genes?

Reply: We apologize for the unclear descriptions about the selection of 11 genes. In this study, we found that the upregulated DE genes were significantly enriched in biological pathway related to glycolysis, while the downregulated DE genes were potentially enriched in lipid metabolism, suggesting the imbalance of glycolysis and lipid metabolism during metastasis. Previous studies showed that the discoordination of glycolysis and lipid metabolism of tumors to be associated with clinical outcome [1]. Therefore, we focused on the upregulated DE genes in the glycolysis and downregulated DE genes in the two enriched lipid metabolism pathways ('Glycerophospholipid metabolism' and 'Ether lipid metabolism'), to explore the metabolic reprogramming phenotypes. Actually, there was a few downregulated DE genes in glycolysis, and upregulated DE genes in the two lipid metabolism pathways, which were not used for the clustering. According to the reviewer's suggestion, we have described this process more clearly in the revised Results section on page 5: "Previous studies showed the discoordination of glycolysis and lipid metabolism of tumors to be associated with clinical outcomes [13]. Therefore, in this study, we focused on the upregulated genes involved in glycolysis and downregulated genes in lipid metabolism to analyze the metabolic reprogramming associated with LUAD metastases. Totally, we extracted the eight upregulated DE genes (*ALDOA*, *ENO1*, *GAPDH*, *GPI*, *LDHA*, *PGAM1*, *PGM2*, *TPII*) from the 'Glycolysis/Gluconeogenesis' pathway and three downregulated DE genes (*PLPPI*, *GPDIL*, *PLD3*) from the two lipid metabolism pathways ('Glycerophospholipid metabolism' and 'Ether lipid metabolism'). Here, the detailed functions of the 11 DE genes are represented in Table S1."

2. The authors identified metabolic phenotypes and their commutations with stromal cells and immune cells using scRNA-seq data. However, details of metabolic pathways and these changes in the three phenotypes are still unknown. The authors additionally need to describe patterns of pathways including genes and metabolites associated with the 11 metabolic genes and the three phenotypes in lung adenocarcinoma.

Reply: In accordance with your suggestion, we have analyzed the association of the mRNA expression of the other glycolytic and lipid metabolic genes in corresponding biological pathways with the three phenotypes. The result showed that 26 glycolytic genes were significantly positively correlated with MPs that gradually increased from MP-I to MP-III through MP-II (Pearson correlation, FDR < 0.05), significantly higher than the number of the negatively correlated genes (Binomial test, $P = 0.0266$). Notably, pyruvate kinase M1/2 (PKM), a key glycolytic enzyme, was gradually increased from MP-I to MP-III, suggesting the stronger activation levels of glycolysis pathway in MP-III. Similarly, 43 lipid metabolic genes were significantly negatively correlated with MPs, significantly higher than the number of positively correlated genes in the two lipid metabolic pathways (Binomial test, $P = 0.0026$). The result showed that there were more dysregulated genes involved in the two kinds of pathways between the three MPs, suggesting the stronger dysregulations of the two kinds of pathways in MP-III. This issue was discussed in the revised Discussion section on page 16: “Additionally, we also found that a lot of other glycolytic and lipid metabolic genes in corresponding biological pathways were significantly positively and negatively correlated with MPs (Pearson correlation, FDR < 0.05, Table S6), such as a key glycolytic enzyme (*PKM*) gradually increasing from MP-I to MP-III. The results indicated the stronger dysregulations of the two kinds of pathways in MP-III. However, whether more glycolytic or lipid metabolic genes could be used to improve the definition of MPs needs further exploration.”

Currently, due to the lack of paired mRNA expression profiles and metabolomics data, we could not estimate the association of metabolites with the 11 metabolic genes and the three phenotypes, which was mentioned in the revised Discussion on page 20: “Although the proteomic provided the evidences of the higher metabolic levels of MP-III, the activities of glycolysis and lipid metabolism still need to be validated by estimating their metabolites.”

3. I recommend that they would add the discussion about master regulators of the changes of the metabolic phenotypes and occurrence of metastasis.

Reply: We thank the reviewer for the helpful comment. We have described the regulators to metabolic reprogramming phenotypes for metastasis in the revised Discussion section on page 19: “The comprehensive network demonstrated that *TP53* mutation, *KEAP1* mutation, and *SETD2* deletion could regulate the mRNA expression of these metabolic genes, bringing about the metabolic reprogramming. Recent studies have shown that mutations in *TP53* [37-39] and *KEAP1* [40] contribute to various metabolic disorders and play pivotal roles in metabolic reprogramming and cancer progression. Furthermore, the protein-protein interaction network from the STRING database (<https://string-db.org/>) indicated that *TP53* could activate glycolysis pathway through up-regulating *PGM2* [37], and hereby enhance the intercellular communication

of MP-III with endothelial cells on the VEGF signaling pathway (*VEGFA-KDR*). Meanwhile, *TP53* could regulate lipid metabolism by interacting with *HIPK4* to *PLD3*. Previous studies have shown that the mutation in *KEAP1* could activate *NRF2*, directing epithelial cells toward metabolic reprogramming of glucose metabolism [40, 41]. In this study, we found that the mutation of *KEAP1* might indirectly interact with a glycolytic gene (*LDHA*) through *COPS5*, which is related to the activation of glycolysis [42]. Meanwhile, *KEAP1* might also affect lipid metabolism through indirectly interaction with *TP53*. The histone H3K36 methyltransferase *SETD2* is frequently mutated or deleted in a variety of human tumors [43-45]. Nevertheless, the role of *SETD2* deletion in promoting metastasis and metabolic reprogramming remains largely undefined. In this study, we found that *SETD2* deletion might participate in glycolysis-lipid metabolism discoordination by indirectly interacting with *TP53* and *KEAP1*.”

4. The authors found that *KEAP1* mutation statuses were associated with the metabolic phenotypes. I think that *KEAP1* is associated with oxidative stress responses and mutations of this gene tend to affect several metabolic pathways. Is there any direct association between *KEAP1* mutations and the detected metabolic changes of glycolysis and lipid metabolism?

Reply: According to the reviewer’s helpful suggestion, we supplemented protein-protein interaction between genomic lesions with the 11 metabolic genes, and found that the mutation of *KEAP1* might indirectly interact with a glycolytic gene (*LDHA*) through *COPS5*, which is related to the activation of glycolysis [2]. Meanwhile, *KEAP1* might also affect lipid metabolism through indirectly interaction with *TP53*. We have described the regulators to metabolic reprogramming phenotypes for metastasis in the revised Discussion section on page 19: “In this study, we found that the mutation of *KEAP1* might indirectly interact with a glycolytic gene (*LDHA*) through *COPS5*, which is related to the activation of glycolysis [42]. Meanwhile, *KEAP1* might also affect lipid metabolism through indirectly interaction with *TP53*.”

5. For intercellular communication, the authors validated expression patterns of 17 genes only using bulk transcriptome datasets of TCGA data. The detected ligand-receptor interaction should be validated using different scRNA-seq datasets of lung adenocarcinoma or by other experimental methods.

Reply: We thank the reviewer for the helpful comment. In according with your suggestion, we additionally validated these cell-cell communications in an independent scRNA-seq data in the revised Results section on page 14: “Collectively, we identified 103 ligand-receptor pairs interacting MP-III with other cell populations (Table S4). Similarly, we identified 154 significant ligand-receptor pairs interacting MP-III with other cell populations in an independent scRNA-seq dataset (GSE123902). Thereinto,

there was 62 overlapped ligand-receptor pairs, significantly higher than expected (Hypergeometric distribution model, $P < 0.0001$, Fig. S4). Notably, mRNA expression of nine ligands expressed on MP-III, including *ANGPTL4*, *VEGFA* etc., were significantly higher than that on MP-I, however, none of them is known driver gene for LUAD. All nine ligands were validated in another scRNA-seq dataset (Wilcoxon rank test, FDR < 0.05, Table S5).”

6. I could not find figure legends.

Reply: We are sorry that the figure legends were not reviewed with the text, which brings greatly confusion for understanding the figures. We have added the figure legends at the end of the revised manuscript.

Minor points;

1. Line 113, p. 7: with with -> with

Reply: Done as suggested.

2. Line 181, pp. 11: I could not find the list of 28 LUAD driver genes.

Reply: We are sorry for lack of gene list. The descriptions of these 28 LUAD driver genes were listed in Table S3.

3. Line 463, pp. 24: the description for pre-processing of gene mutation data was insufficient.

Reply: We are sorry for unclear description of pre-processing of gene mutation. For somatic mutation data, we used Mutect2 somatic variant calls in mutation annotation format from the cancer genome atlas (TCGA) data portal. We constructed a discrete mutation profile where the rows were genes and the columns were samples, which was marked 1 if the gene was nonsynonymous mutation and 0 if it was not. We have replenished the description of mutation profile in the revised Methods section on page 23: “For somatic mutation data derived from the Illumina Genome Analyzer DNA Sequencing GAIIX platform, only the nonsynonymous mutations were included, and a discrete mutation profile (mutated or not) including 17,821 genes was generated using Mutect2 somatic variant calls in mutation annotation format.”

References

1. Karasinska JM, Topham JT, Kalloger SE, et al. Altered Gene Expression along the Glycolysis-Cholesterol Synthesis Axis Is Associated with Outcome in Pancreatic Cancer. *Clin Cancer Res.* 2020;26(1):135-46.
2. Li P, Gao L, Cui T, et al. Cops5 safeguards genomic stability of embryonic stem cells through regulating cellular metabolism and DNA repair. *Proc Natl Acad Sci U S A.* 2020;117(5):2519-25.

REVIEWERS' COMMENTS:

Reviewer #1 (Remarks to the Author):

The authors significantly improved the quality of their manuscript by addressing all my concerns.

I only suggest minor english corrections, such as:

-line 169: " a certain percentage" is not a very scientific way to describe a concept

- lines 227-228: it should read the lowEST and the highEST

- line 274: there WERE (not there was)

Please review the english part accordingly throughout the entire manuscript.

Other than that for me it is OK to accept the manuscript.

Reviewer #2 (Remarks to the Author):

The authors have responded appropriately to all my previous comments and I now am in favor of acceptance

Reviewer #3 (Remarks to the Author):

The authors have addressed the concerns in the revised manuscript. To improve the manuscript, I just have recommendations for figure legends as below;

1. In the legend of Figure 7, I think that the short description of hierarchical plots should be there in addition to the reference of the original paper.
2. I think that Table S2 is helpful to understand the relationship of statistical significance and each of the factors in Figure 2. Please refer to Table S2 in the legend of Figure 2 if possible.

Reviewers' comments:

Reviewer #1 (Remarks to the Author):

Line 169: " a certain percentage" is not a very scientific way to describe a concept

Reply: We are sorry for the wrong description. We have rephrased it in the revised Results section in Lines 144: “The above result showed that a percentage of early-stage patients without any metastasis could be clustered into MP-III; thus, we inferred that these patients might have high-risk of metastasis, resulting in poor prognoses.”

Lines 227-228: it should read the lowEST and the highEST

Reply: Done as suggested.

Line 274: there WERE (not there was)

Reply: Done as suggested.

Please review the english part accordingly throughout the entire manuscript.

Reply: We thank the reviewer for this suggestion. We have reviewed the entire manuscript.

Reviewer #3 (Remarks to the Author):

In the legend of Figure 7, I think that the short description of hierarchical plots should be there in addition to the reference of the original paper.

Reply: We thank the reviewer for the insightful suggestion. In according with your suggestion, we have added the description of hierarchical plots in the legend of Figure 7 on Page 34: “(a) Hierarchical plot showing the interactions between the MPs and other cells in the ANGPTL signaling pathway. This plot consists of two parts: Left and right portions highlight the autocrine and paracrine to MPs and to other cells, respectively. Solid and open circles represent source and target, respectively. Circle sizes are proportional to the number of cells in each cell group and edge width represents the

communication probability. Edge colors are consistent with the signaling source.”

I think that Table S2 is helpful to understand the relationship of statistical significance and each of the factors in Figure 2. Please refer to Table S2 in the legend of Figure 2 if possible.

Reply: Thank you for the insightful suggestion. According to your suggestion, we have revised the legend of Figure 2 on Page 32: “Circle heatmap of clinical information (Stage, Age, Gender, Transcriptomics subtypes, Molecular scores, and TMB) and oncogenic events (*TP53*, *KEAP1*, and *SETD2*). The significance of differences (*P* value) among the three MPs, estimated by Fisher’s exact test, is shown after each characteristic. **The detailed numbers of each characteristic in the MPs are presented in Supplementary Table 2. Radar chart (inner) displays four molecular scores and TMB of the three MPs.”**